# Investigation of inherited noncoding genetic variation impacting the pharmacogenomics of childhood acute lymphoblastic leukemia treatment

Kashi Raj Bhattarai [1,2,15], Robert J. Mobley [1,2,15], Kelly R. Barnett[1,2], Daniel C. Ferguson[1,2], Baranda S. Hansen [3,4], Jonathan D. Diedrich [1,2], Brennan P. Bergeron [1,2,5], Satoshi Yoshimura[1,2,6], Wenjian Yang [1,2], Kristine R. Crews [1,2], Christopher S. Manring[7], Elias Jabbour[8], Elisabeth Paietta[9], Mark R. Litzow [10], Steven M. Kornblau[8], Wendy Stock[11], Hiroto Inaba [1,12], Sima Jeha[1,12], Ching-Hon Pui [1,12], Cheng Cheng[13], Shondra M. Pruett-Miller [3,4], Mary V. Relling [1,2], Jun J. Yang [1,2,5,14], William E. Evans [1,2] & Daniel Savic [1,2,5,14] ✉

Defining genetic factors impacting chemotherapy failure can help to better predict response and identify drug resistance mechanisms. However, there is limited understanding of the contribution of inherited noncoding genetic variation on inter-individual differences in chemotherapy response in childhood acute lymphoblastic leukemia (ALL). Here we map inherited noncoding variants associated with treatment outcome and/or chemotherapeutic drug resistance to ALL *cis*-regulatory elements and investigate their gene regulatory potential and target gene connectivity using massively parallel reporter assays and three-dimensional chromatin looping assays, respectively. We identify 54 variants with transcriptional effects and high-confidence gene connectivity. Additionally, functional interrogation of the top variant, *rs1247117*, reveals changes in chromatin accessibility, PU.1 binding affinity and gene expression, and deletion of the genomic interval containing *rs1247117* sensitizes cells to vincristine. Together, these data demonstrate that noncoding regulatory variants associated with diverse pharmacological traits harbor significant effects on allele-specific transcriptional activity and impact sensitivity to antileukemic agents.

Due to continual advances in treatment protocol over the last 60 years, the survival rate of the most common malignancy in children, acute lymphoblastic leukemia (ALL), has dramatically improved to over 90% in high-income countries[1]. Despite these advances, survival rates of pediatric patients experiencing refractory or relapsed ALL were only 30–50%, and those of adults were especially low (-10%)[2]. Thus,

improving the understanding of the underlying genetic risk factors impacting response to ALL chemotherapy is a major step in improving outcomes for patients with refractory or relapsed ALL.

Genome-wide association studies (GWAS) have identified numerous inherited DNA sequence variants associated with treatment outcome in childhood ALL from clinical trials carried out by St. Jude

---

Children's Research Hospital and the Children's Oncology group[3–5]. This includes GWAS analyses that identified inherited genetic contributors associated with patient relapse[4,5] and persistence of minimal residual disease (MRD) after induction chemotherapy[3], which is an early indicator of treatment failure[6–9]. In addition, ex vivo chemotherapeutic drug sensitivity testing using primary ALL cells from patients serves as an informative pharmacological phenotype[10]. When integrated with genotype profiling for GWAS, these analyses identify variants contributing to antileukemic drug resistance that reflects in vivo and ex vivo resistance and is, therefore, predictive of treatment outcomes in patients[10–22].

Because most GWAS variants, including pharmacogenomic variants[23,24], lie in noncoding sequences in the human genome, their connection to gene regulation and cellular biology has yet to be established. Moreover, given that dozens of variants are typically in

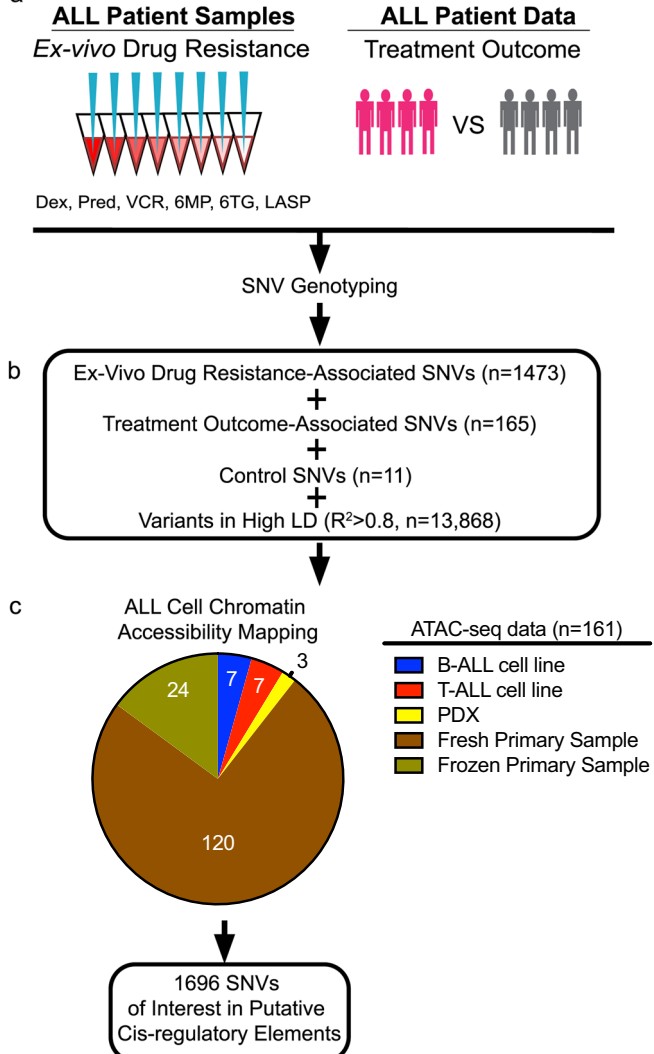

**Fig. 1 | Identification and mapping of regulatory variants impacting the pharmacogenomics of ALL treatment. a** SNVs of interest from GWAS were pursued based on association with ex vivo chemotherapeutic drug resistance in primary ALL cells from patients and/or treatment outcome. *Dex* dexamethasone, *Pred* prednisolone, *VCR* vincristine, *6MP* 6-mercaptopurine, *6TG* 6-thioguanine, *LASP* L-asparaginase. **b** GWAS SNVs were combined with ALL disease susceptibly control GWAS SNVs and SNVs in high LD ($R^2 > 0.8$) and **c** mapped to accessible chromatin sites in ALL cell lines, ALL PDXs, and primary ALL cells from patients. Of the 1696 SNVs mapped to accessible chromatin sites, 35 are control SNVs. Source data are provided in the Source Data file.

strong linkage disequilibrium (LD) with an associated sentinel variant, pinpointing causal variants at GWAS loci has been challenging. Noncoding GWAS variants have been consistently linked to disruption of cis-regulatory element (CRE) activity and gene regulation[25]. As a result, the functional evaluation of these regulatory variants involves an examination of their allele-specific activities on transcriptional output, which has traditionally been a low-throughput endeavor. Therefore, the functional investigation of all associated regulatory variations at GWAS loci (sentinel and LD) proved to be an intractable hurdle to investigators. Recent technological advances, however, have ameliorated these challenges through the advent of massively parallel reporter assays (MPRAs) where the reporter aspect is often a self-transcribed barcode in the 3′-UTR of a reporter gene that is detected using next-generation sequencing. MPRAs allow the simultaneous, rapid, and robust detection of differences in transcriptional output from a library of cis-regulatory sequences of interest[26–28]. MPRAs have since been applied to the study of regulatory variation at GWAS loci through an examination of allele-specific effects on reporter gene expression[29–35].

Another challenge is connecting regulatory variation at promoter-distal CREs to a target gene, as the closest gene may not be the target gene[36]. To circumvent these challenges, regulatory variation can be coupled to transcriptomics to identify variants impacting the expression of a candidate target gene through expression quantitative trait locus (eQTL) mapping[36,37]. Functional genomics offers additional solutions through the mapping of three-dimensional (3D) genomic interactions[38]. Because promoter-distal CREs (e.g., enhancers) regulate gene expression through long-range 3D looping to the promoters of target genes[25], an attractive assay to identify gene targets of promoter-distal regulatory variants is promoter capture Hi-C (promoter CHiC)[39]. Promoter capture Hi-C, and related 3D chromatin interaction assays have been implemented at multiple GWAS loci to identify gene targets of promoter-distal regulatory variation[29,36,40–43].

In this work, to better understand the underlying genetic and gene regulatory factors that impact diverse pharmacological traits in ALL, we performed a comprehensive functional interrogation of GWAS regulatory variants that map to ALL accessible chromatin sites and that are associated with ALL treatment outcome (i.e., relapse and persistence of MRD after induction chemotherapy) in patients and/or ex vivo chemotherapeutic drug resistance in primary ALL cells from patients using MPRA. We coupled these results with promoter CHiC to identify candidate target genes of functional regulatory variants with significant allele-specific effects on reporter gene expression. Finally, we functionally investigated the impact of the top regulatory variants chemotherapeutic drug resistance in ALL cell lines. Collectively, this study implements a comprehensive functional investigation of the allele-specific, gene-regulatory effects of noncoding variants associated with the pharmacogenomics of ALL treatment, and, therefore, fills an unmet need for large-scale functional examinations of regulatory GWAS variants associated with ALL pharmacological traits.

## Results
### Identification of noncoding regulatory variants impacting the pharmacogenomics of ALL treatment
Single-nucleotide variants (SNVs) impacting diverse pharmacological traits in ALL were identified for functional interrogation. We chose SNVs associated with relapse or persistence of MRD after induction chemotherapy in childhood ALL patients to investigate the role of inherited noncoding regulatory variants impacting clinical phenotypes (i.e., treatment outcome). These SNVs were identified from published GWAS of ALL patients enrolled in St. Jude Children's Research Hospital and the Children's Oncology Group clinical protocols[3–5] (see Methods for variant selection criteria). Variant selection also included prioritization for treatment outcome SNVs associated with drug resistance phenotypes in primary ALL cells to enrich for variation impacting ALL

cell biology (see Methods for variant selection criteria). These treatment outcome-associated variants, as well as all variants in high LD ($r^2 > 0.8$) with the sentinel GWAS variants, were further evaluated (Fig. 1a, b).

We also identified variants directly associated with ex vivo chemotherapeutic drug resistance in primary ALL cells from patients by performing GWAS analyses using SNV genotype information and ex vivo drug resistance assay results for six antileukemic agents (prednisolone, dexamethasone, vincristine, L-asparaginase, 6-mercaptopurine [6MP] and 6-thioguanine [6TG]) in primary ALL cells from 312–344 patients (not all patients were tested for all drugs) enrolled in the Total Therapy XVI clinical protocol at St. Jude Children's Research Hospital (see Methods). We further prioritized functional ex vivo drug resistance SNVs by determining if they were eQTLs in primary ALL cells or related cell types (i.e., whole blood and EBV-transformed lymphocytes) from the Genotype-Tissue Expression (GTEx) consortium[37] (see Methods for variant selection criteria). Ex vivo drug resistance variants that were also identified as eQTLs, as well as variants in high LD ($r^2 > 0.8$) with these sentinel GWAS variants, were further evaluated (Fig. 1a, b).

GWAS have also been performed for childhood ALL disease susceptibility and identified several GWAS loci harboring variants with genome-wide significance[44–50]. Several follow-up studies of these GWAS loci have identified candidate causal noncoding variants and mechanisms involving gene regulatory disruptions[51–53]. As a result, we used ALL disease susceptibility variants ($n = 11$), as well as variants in high LD ($r^2 > 0.8$) with them, for further analysis as positive controls in our study (Fig. 1a, b).

Because most of these variants map to noncoding portions of the human genome, these data point to disruptions in gene regulation as the underlying mechanism of how these variants impact ALL cell biology. We therefore utilized assay for transposase-accessible chromatin with high-throughput sequencing (ATAC-seq)[54] chromatin accessibility data in 161 ALL cell models, comprised of primary ALL cells (cryopreserved, $n = 24$[55]; fresh, $n = 120$[56]), ALL cell lines ($n = 14$) and ALL patient-derived xenografts (PDXs, $n = 3$), to uncover which variants map to putative CREs in ALL cells[57] (i.e., regulatory variants; Fig. 1c). Although we detected variation in ATAC-seq TSS enrichment scores and peak counts that is to be expected from such a large, mixed cohort of ALL cell models, the peaks called were largely reproducible (found in >3 samples) within each group (Supplementary Fig. 1a–c). ATAC-seq data from primary ALL cells, ALL cell lines, and PDXs were combined and identified 1696 regulatory variants at accessible chromatin sites in ALL cells for functional investigation (Fig. 1c and Supplementary Data 1).

## Assessing the impact of regulatory variation on transcriptional output using MPRA

To examine the functional effects of these 1696 regulatory variants on transcriptional output in a high-throughput manner we utilized a barcode-based MPRA platform[29,32] to measure differences in allele-specific transcriptional output (Fig. 2a). Oligonucleotides containing 175-bp of genomic sequence centered on each reference (ref) or alternative (alt) variant allele, a restriction site, and a unique 10-bp barcode sequence were cloned into plasmids. An open reading frame containing a minimal promoter driving GFP was then inserted at the restriction site between the alleles of interest and their unique barcodes (Fig. 2a). We utilized 28 unique 3'UTR DNA barcodes per variant allele (56 barcodes per regulatory variant), and variants near bidirectional promoters (47 total variants) were tested using both sequence orientations. In total, 97,608 variant-harboring oligonucleotides were evaluated for allele-specific differences in gene regulatory activity (Fig. 2a).

Following transfection into 7 different B-cell precursor ALL (B-ALL; 697, BALL1, Nalm6, REH, RS411, SEM, SUPB15) and 3 T-cell ALL (T-ALL; CEM, Jurkat, P12-Ichikawa) human cell lines ($n = 4$

transfections per cell line; 40 total), the transcriptional activity of each allele variant was measured by high-throughput sequencing to determine the barcode representation in reporter mRNA and compared to DNA counts obtained from high-throughput sequencing of the MPRA plasmid pool (Fig. 2a). In the 10 cell lines MPRA detected 4633 instances of significant differential activity between alleles across 91% (1538/1696) of regulatory variants tested (Fig. 2b, c, Supplementary Data 2). The 10 ALL cell lines showed substantial differences in the total number of regulatory variants harboring significant allele-specific activity, which we suspect largely stems from differences in transfection efficiency (Fig. 2c). Importantly, when comparing changes in allele-specific MPRA activity for each regulatory variant we found that significant changes in activity (adj. $p < 0.05$) were highly correlated between ALL cell lines, with 87% concordance in allelic-specific activity, suggesting that significant MPRA hits were likely to be robust and reproducible between cell lines (Fig. 2d). Allele-specific MPRA activities were also correlated using all pairwise cell line comparisons for each regulatory variant, irrespective of significance (Supplementary Fig. 2a). Importantly, 31 of the 35 positive control variants (i.e., ALL disease susceptibility-associated variants and variants in high LD) showed significant allelic effects in at least 1 cell line, and 10 showed significant and concordant allelic effects in at least three ALL cell lines, including two variants (rs3824662 at GATA3 locus and rs75777619 at 8q24.21) directly associated with ALL susceptibility[44,49,52] (Supplementary Data 2). The risk A allele at rs3824662 was associated with higher GATA3 expression and chromatin accessibility and demonstrated significantly higher allele-specific activity in our MPRA[44,52], thereby demonstrating that the MPRA could detect allelic effects previously identified by others.

To further validate MPRA hits in an ex vivo model, we performed MPRA using two B-ALL PDX samples that were freshly harvested from mice. These samples detected 26 and 67 significant gene regulatory variants, respectively, and showed significant correlation with the cell line MPRA data (Supplementary Fig. 2b, c, Supplementary Data 3). We attribute the detection of relatively lower numbers of variants in PDXs to technical effects stemming from poor transfection efficiency and limited cell survival ex vivo. Overall, our data suggest that the cohort of SNVs tested contained functional regulatory variants with the potential to impact gene regulation.

## Identification of functional regulatory variants showing reproducible and concordant changes in allele-specific gene expression

To further focus on regulatory variants most likely to broadly impact gene regulation in ALL cells, we prioritized 556 variants with significant (adj. $p < 0.05$) and concordant allele-specific activities in at least three ALL cell lines (i.e., functional regulatory variants; Fig. 3a–d, Supplementary Data 4). Most of these functional regulatory variants (318/556) mapped to accessible chromatin found only in primary ALL cell samples, underscoring the importance of incorporating chromatin architecture from primary ALL cells, and 54 functional regulatory variants mapped to transcription factor footprints in primary ALL cells (Supplementary Fig. 3). Additionally, we used Genomic Regions Enrichment of Annotations Tool (GREAT) to associate these SNVs with their nearby genes and search for enrichment in gene ontology biological processes pathways[58]. Although GREAT identified gene associations for nearly all SNVs, we found no significant pathway associations (Supplementary Data 4 and 5). Because further functional investigation of variants in primary ALL cells or PDXs ex vivo is largely intractable, we focused on 210 functional regulatory variants that were detected in open chromatin in one of the 14 ALL cell lines that we had generated ATAC-seq data (Fig. 3d). Most of these variants (159/210; 76%) were also found in accessible chromatin in PDXs and/or in primary ALL cells from patients (Fig. 3d).

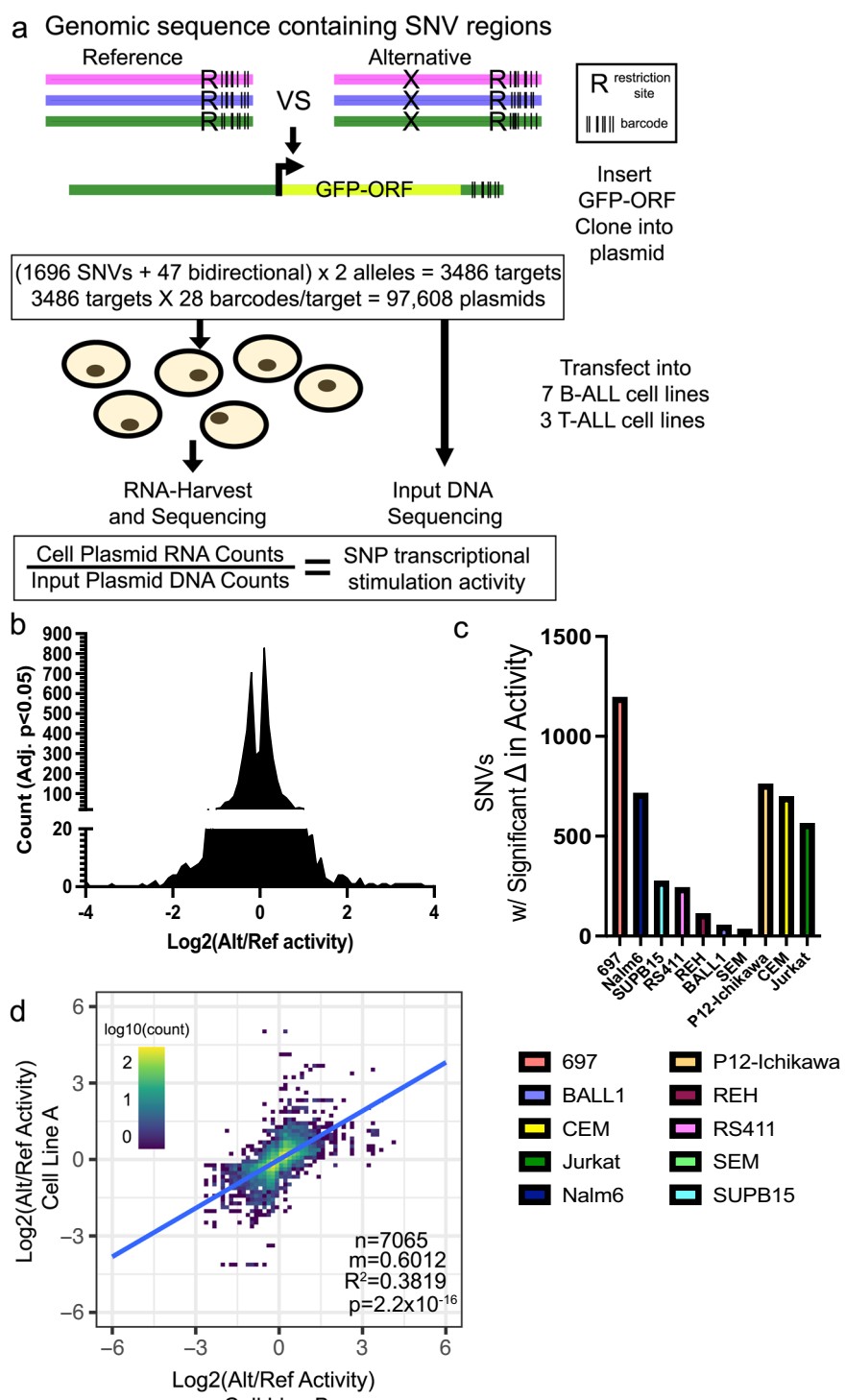

**Fig. 2 | MPRA identifies regulatory variants with allele-specific effects on gene expression. a** Diagram describing design of MPRA (also see Methods). **b**–**d** Significant MPRA hits were identified by Benjamini–Hochberg FDR corrected two-tailed Students $T$ tests. **b** Distribution of significant changes in allele-specific transcriptional activity across all SNVs. **c** Number of MPRA SNVs showing significant (Adj. $p < 0.05$) changes in allele-specific transcriptional activity in each ALL cell line. **d** Pairwise linear correlation between changes in allele-specific transcriptional activity for all significant (Adj. $p < 0.05$) changes across all cell lines. $R^2$ correlation and $p$ value are provided. All source data and statistical parameters are provided in the Source Data file.

For additional validation using traditional luciferase reporter assays, we prioritized these 210 functional regulatory variants based on allele-specific effect size and selected high-ranking SNVs. Dual-luciferase reporter assays showed similar allele-specific changes in activity to that which was detected by MPRA for 7 SNVs tested (Supplementary Fig. 4a–k). In fact, a significant positive correlation ($p = 0.0017$) was observed between the allelic effects detected by MPRA and luciferase reporter assays (Supplementary Fig. 4l). Together, these analyses assessed the robustness of our MPRA screen of functional regulatory variants and identified 556 SNVs with reproducible and concordant allele-specific effects on gene regulation. Importantly, 210 of the 556 significant hits that were concordant in at

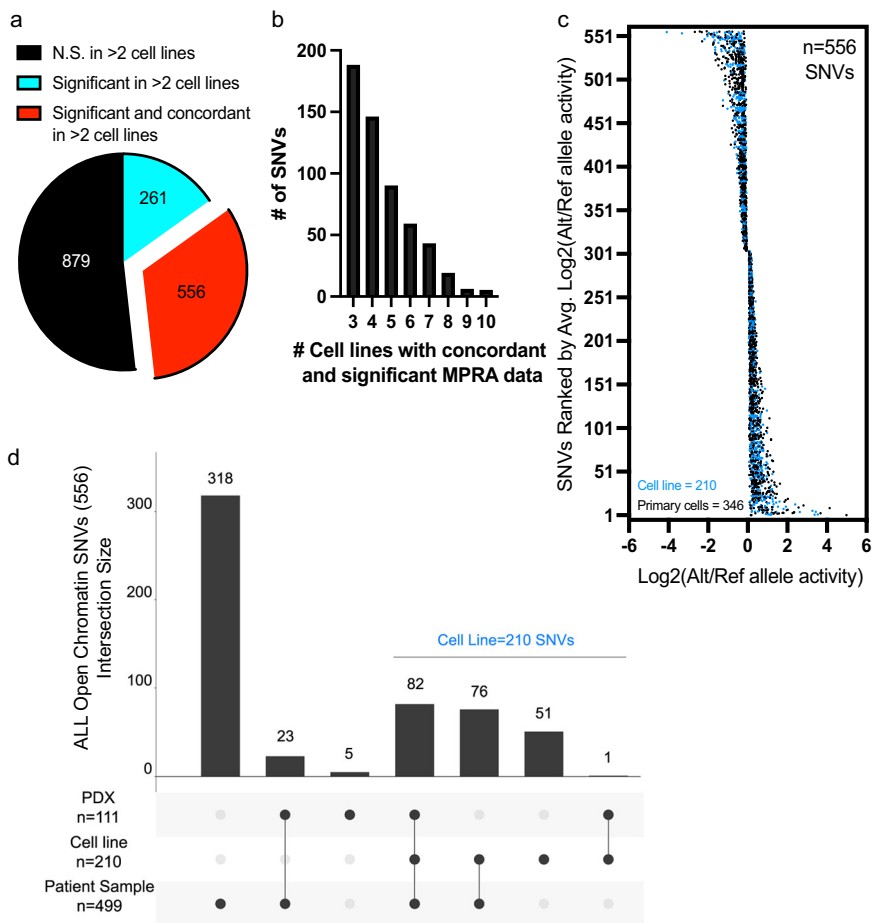

**Fig. 3 | Identification of functional regulatory variants with reproducible and concordant effects in allele-specific stimulation of transcriptional activity.**
**a** 556 of the 1696 SNVs assayed are functional regulatory variants with reproducible (FDR < 0.05 in >2 cell lines) and concordant (same directionality in >2 cell lines) changes in allele-specific activity. **b** Frequency distribution plot showing the number of cell samples showing concordant and significant MPRA activity of variants. **c** Plot showing the distribution of log$_2$-adjusted activity between alternative

(Alt) and reference (Ref) alleles across 556 functional regulatory variants. 210 SNVs (in blue) mapped to accessible chromatin sites in ALL cell lines and 346 SNVs (in black) mapped only to accessible chromatin sites identified in primary ALL cells and/or PDXs. **d** Upset plot shows how many functional regulatory variants map to open chromatin in diverse ALL cell models. 210 of the 556 functional regulatory variants are found in accessible chromatin sites that were identified in an ALL cell line. Source data are provided in the Source Data file.

least three cell lines were found in open chromatin sites in ALL cell lines and, therefore, warranted further exploration.

## Association of functional regulatory variants with putative gene targets

To better understand how these variants impact cellular phenotypes, we first determined if the 210 functional regulatory variants found in accessible chromatin sites in ALL cell lines could be directly associated with a target gene. While 35 functional regulatory variants were localized close (±2.5 kb) to nearby promoters (Fig. 4a, Supplementary Data 4 and 6), 175 variants were promoter-distal (>2.5 kb), and therefore likely to map to CREs with unclear gene targets (Fig. 4a). While CREs are often associated with the nearest genes, 3D chromatin looping methods are a more reliable method to associate a CRE with its target gene promoter. In pursuit of evidence-based association of promoters and specific CREs, we performed two related chromatin looping methods, H3K27Ac HiChIP[59] and promoter capture HiC (CHiC)[39], in 8 of 10 ALL cell lines used in MPRA and determined that 19 of the 175 non-promoter functional regulatory variants showed connectivity to distal promoters in the same cell line where allele-specific MPRA activity and chromatin accessibility were detected (Fig. 4a, Supplementary Data 6). Interestingly, H3K27Ac HiChIP and promoter CHiC called similar numbers of loops across all 8 cell lines (690,579

versus 660,313, respectively), but promoter CHiC loop calling was more consistent per cell line (Supplementary Fig. 5, Supplementary Data 7). HiChIP detected no looping at any of the 556 reproducible and concordant SNVs from the MPRA, and the 19 SNVs showing connectivity to a promoter were solely detected by promoter CHiC, further highlighting the utility of this method in GWAS-oriented studies[41,60–63].

In prioritizing functional regulatory variants, we were interested in the gene regulatory impact of variants at TSS-proximal promoter-associated versus TSS-distal promoter-connected CREs as measured by MPRA. Interestingly, we found that SNVs found at TSS-distal open chromatin sites, promoter-associated or not, showed higher allele-specific changes in MPRA activity than those at promoters (Fig. 4b). While we acknowledge that many of the 156 variants for which we did not detect a relationship with a promoter are likely to have meaningful gene targets, we focused on CREs containing variants with known gene targets in ALL cells for functional validation. Amongst the TSS-distal promoter-connected functional regulatory variants, we found that distal intergenic and intronic SNVs showed significantly higher allele-specific activity than those in UTRs (Fig. 4c). These data suggest that the most robust allelic effects attributable to these regulatory variants are likely to occur at distal intergenic and intronic sites >2.5 kb from the TSS of the target gene.

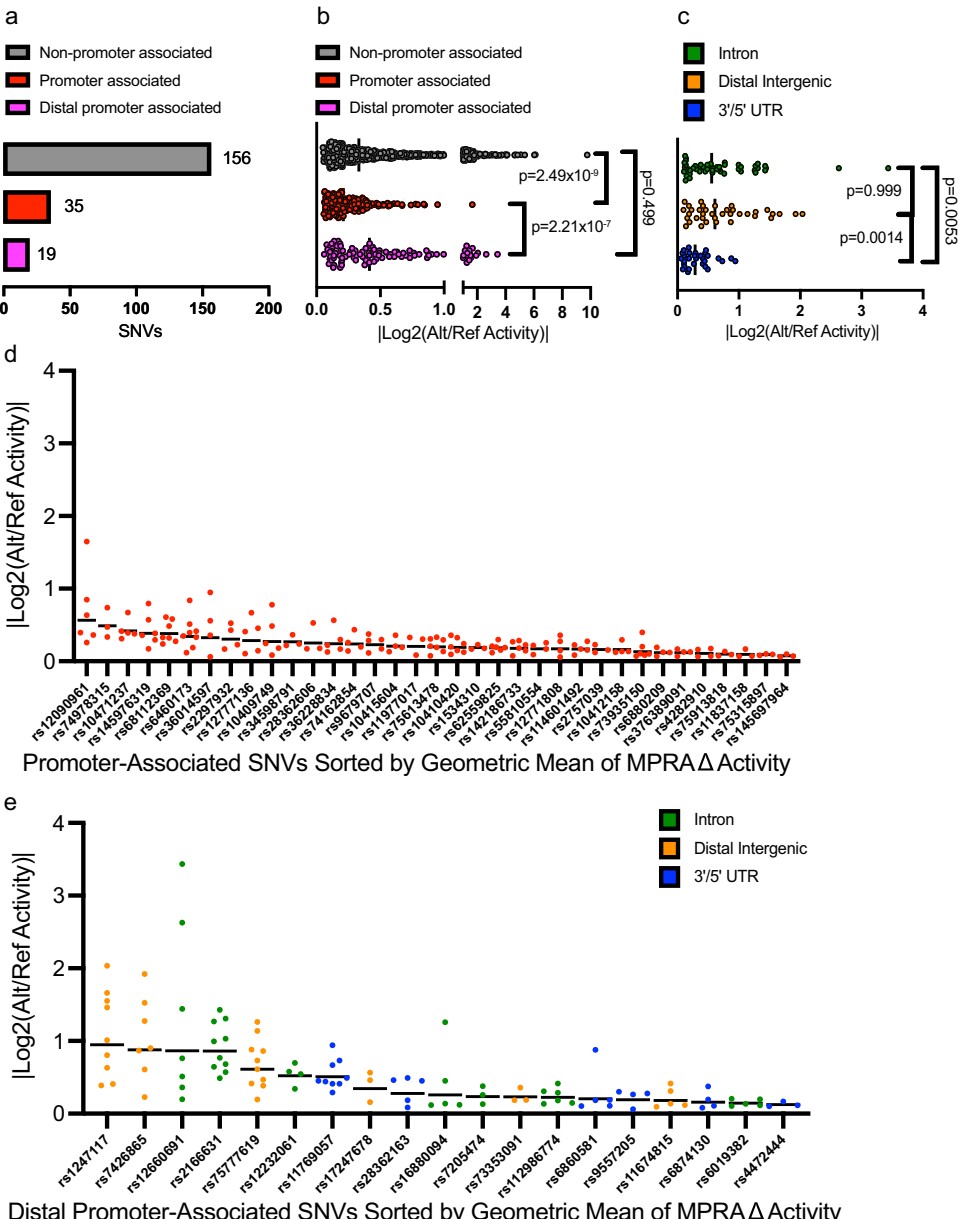

**Fig. 4 | Promoter CHiC identifies target genes of functional regulatory variants.**
**a** Data show the number of functional regulatory variants mapping to open chromatin in cell lines that associate directly with promoters (within 2.5 kb) or that are distally promoter-connected via promoter CHiC. **b** MPRA data show distal regulatory variants in accessible chromatin (some promoter-connected by promoter CHiC data) exhibit stronger effects on allele-specific activity than promoter-associated functional regulatory variants. ANOVA with Kruskal–Wallis test was performed with Dunn's correction for multiple comparisons. **c** Amongst distally promoter-connected functional regulatory, variants that map to intronic and distal intergenic sequences showed greater activity than those in UTRs. ANOVA with Kruskal–Wallis test was performed with Dunn's correction for multiple comparisons. **d**, **e** Data show the ranked allele-specific activity distribution of MPRA data for **d** promoter-associated functional regulatory variants and **e** distally promoter-connected functional regulatory variants. All source data and statistical parameters are provided in the Source Data file.

Next, we ranked TSS-proximal promoter-associated and TSS-distal promoter-connected functional regulatory variants by the geometric mean of their significant MPRA data to account for the magnitude of allele-specific activity and the reproducibility of a significant change across ALL cell lines (Fig. 4d, e). This analysis identified *rs1247117* as the most robust functional regulatory variants, which we then pursued for mechanistic understanding (Fig. 4e).

### rs1247117 determines genomic accessibility, PU.1 binding, and EIF3A expression
We pursued functional validation of *rs1247117* based on its highest-ranking geometric mean of MPRA allelic effect. *rs1247117* is in high LD

with two GWAS sentinel variants (rs1312895, $r^2 = 0.99$; *rs1247118*, $r^2 = 1$) that are associated with persistence of MRD after induction chemotherapy[3]. This functional regulatory variant maps to a distal intergenic region harboring chromatin accessibility downstream of the *CACUL1* gene, for which it is an eQTL in EBV-transformed lymphocytes[37]. However, we found that *rs1247117* loops to the *EIF3A* promoter in Nalm6 B-ALL cells (Fig. 5a). We, therefore, explored how this accessible chromatin site might recruit transcriptional regulators that would depend on the allele present at *rs1247117*. For this, we first performed ChIP-seq for RNA pol II and H3K27Ac, which confirmed RNA Pol II occupancy and H3K27Ac enrichment in Nalm6 cells, indicating that *rs1247117* is associated with an active CRE (Fig. 5a). Through an

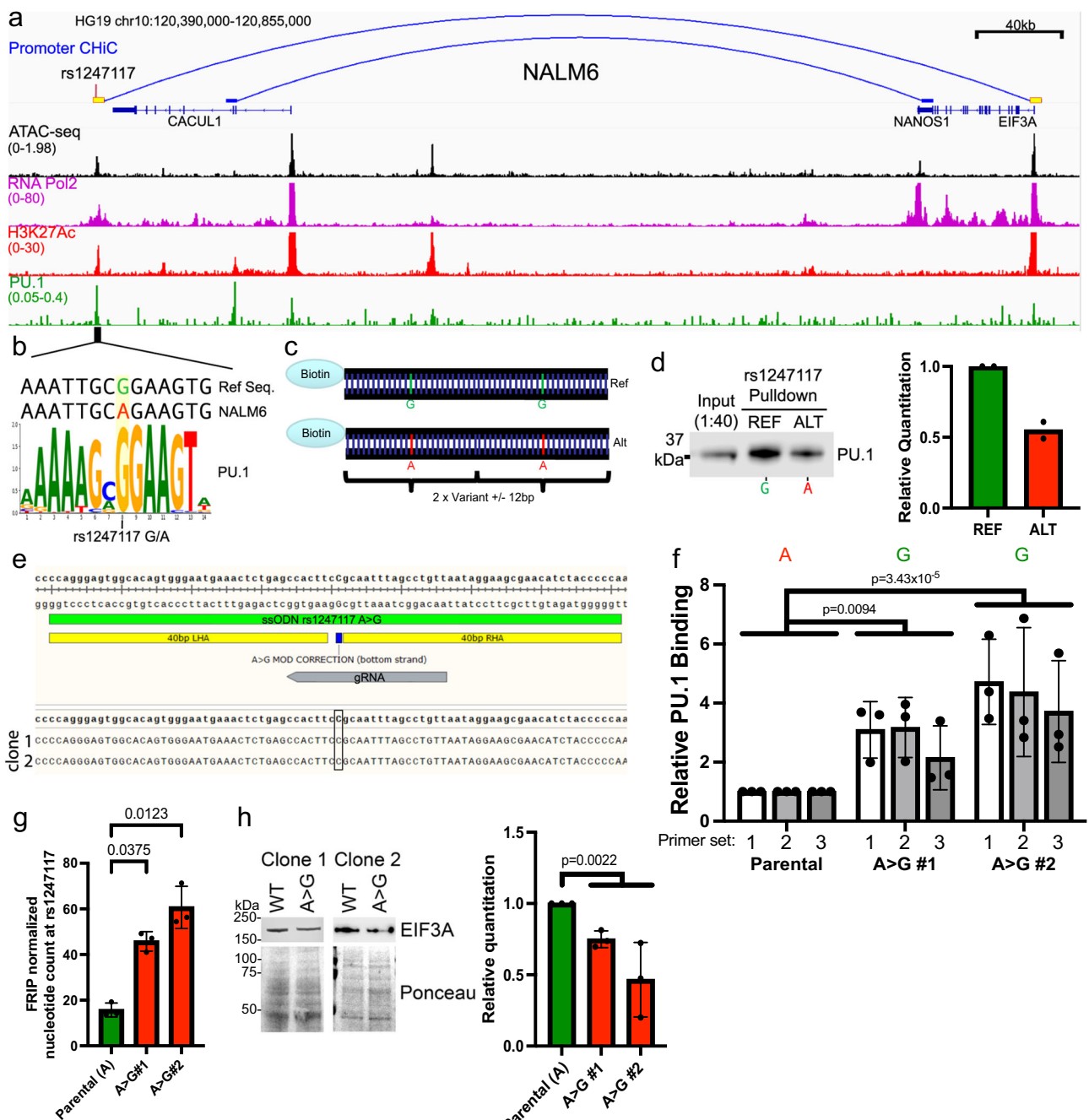

**Fig. 5 | rs1247117 impacts PU.1 occupancy and EIF3A expression. a** IGV genome browser image in Nalm6 cells showing the genomic context, chromatin accessibility, and *EIF3A* promoter connectivity using promoter CHiC of the top functional regulatory variant, rs1247117, with the highest allele-specific MPRA activity. Genomic binding profiles are also shown for RNA polymerase II (RNA Pol2), histone H3 lysine 27 acetylation (H3K27Ac), and PU.1. **b** rs1247117 lies in a PU.1 binding motif. The human genome reference sequence, Nalm6 genome sequence, location of rs1247117, and PU.1 position weight matrix are shown. **c** Design of biotinylated DNA probes for in vitro rs1247117 pulldown. **d** Biotinylated DNA pulldown shows rs1247117 allele-dependent enrichment of PU.1 binding. Blot shown is representative of two independent experiments. Densitometric quantification of two blots is shown. **e** CRISPR/Cas9 was used to change the allele at rs1247117 from A > G in Nalm6 cells. Data show the location of gRNA and ssODN, as well as NGS reads

obtained from clone 1 and 2 at rs1247117. **f** PU.1 ChIP-PCR shows increased PU.1 binding at the rs1247117 locus using two A > G modified clones and 3 primer sets. Data shown are mean ± SD of three independent experiments for each primer set. Two-way ANOVA with Dunnett's multiple comparisons correction, *n* = 3. **g** ATAC-seq data normalized for frequency of reads in peaks (FRIP) show a significantly higher count of G nucleotides in two clones of A > G modified Nalm6 cells compared to the count of A nucleotides detected in parental Nalm6 cells. Data shown are the mean ± SD. Bonferroni corrected, two-tailed Student's *T* tests, *n* = 3. **h** Western blots and quantification showing decreased EIF3A expression in A > G modified Nalm6 cells. Blots shown are representative of three independent experiments. Quantification data shown are the mean ± SD. Two-tailed Student's *T* tests compare parental Nalm6 to combined data from A > G clones, *n* = 3. All source data and statistical parameters are provided in the Source Data file.

examination of the underlying DNA sequence spanning *rs1247117*, we found that the reference guanine (G) risk allele at rs1247117 resides in a PU.1 transcription factor binding motif that is disrupted by the alternative adenine (A) allele (Fig. 5b). Although the risk G allele is the reference allele, the alternative A allele is more common in human populations. Supporting PU.1 binding at this location, accessible chromatin profiling in primary ALL cells identified an accessible chromatin site and PU.1 footprint spanning *rs1247117* in diverse ALL samples (Supplementary Fig. 6a, b). Significantly greater chromatin accessibility at *rs1247117* was also observed in heterozygous (GA) patient samples compared to patient samples homozygous for the alternative A allele (Supplementary Fig. 6c), and the G allele at *rs1247117* harbored significantly greater ATAC-seq read count compared to the A allele (Supplementary Fig. 6d). Importantly, we determined that PU.1 was bound at this site in Nalm6 cells using CUT and RUN[64] (Fig. 5a).

Nalm6 cells contain the alternative A allele that disrupts the PU.1 motif at *rs1247117*, yet our data suggests that this site still binds PU.1 (Fig. 5a, b). This led us to hypothesize that PU.1 binding affinity for the PU.1 motif surrounding *rs1247117* would be strengthened by the risk G allele. Therefore, we designed biotinylated DNA probes containing two tandem 25-bp regions centered on reference G or alternative A allele-containing *rs1247117* to test this hypothesis (Fig. 5c). Using biotinylated probes, we performed an in vitro DNA-affinity pulldown from Nalm6 nuclear lysate and found that while PU.1 was indeed bound to the alternative A allele, PU.1 was more robustly bound to the reference G allele at *rs1247117* (Fig. 5d). To further assess the impact of the *rs1247117* allele on PU.1 binding, we changed the Nalm6 allele from A to G using CRISPR/Cas9 (Fig. 5e; AA = parental genotype, GG = mutated genotype). We used ChIP-PCR to determine that PU.1 binding was increased with the G allele relative to the A allele at the CRE containing *rs1247117* in two A > G Nalm6 clones across 3 unique primer sets within the PU.1 peak at rs1247117 that was detected in Nalm6 cells (Fig. 5f). We then asked if transposase accessibility was also increased at the CRE containing *rs1247117* when the G allele was present. Using ATAC-seq, we found that accessibility was indeed increased at *rs1247117* in mutated Nalm6 cells with the G allele when compared to the parental Nalm6 cells containing the A allele (Fig. 5g). These data suggest that the risk G allele increases genomic accessibility and the affinity of PU.1 binding at *rs1247117* relative to the alternative A allele.

We were next interested in how allele-specific PU.1 binding at *rs1247117* was related to the expression of the protein encoded by the connected gene, EIF3A. We found that the G allele, which increased recruitment of PU.1, resulted in decreased expression of EIF3A when compared to Nalm6 cells with the A allele (Fig. 5h). These data suggest that PU.1 recruitment to the CRE containing *rs1247117* results in a net-repressive effect on EIF3A protein levels, and that less PU.1 recruitment with the A allele results in greater EIF3A expression.

### Deletion of CREs containing top MPRA SNVs demonstrates their impact on drug sensitivity

Clonal selection can lead to the accumulation of random SNVs and even larger structural variations[65] that can confound functional interpretation of more complex *trans* phenotypic effects. Therefore, to examine the connection between *rs1247117* and the persistence of MRD after induction chemotherapy, we decided to use CRISPR/Cas9 to delete the CRE containing *rs1247117* in heterogeneous cell pools of Nalm6 and SUPB15 cells (rs1247117 del) to avoid clonal selection (Fig. 6a, b, Supplementary Fig. 7a). Given that loss of the CRE containing *rs1247117* would abolish PU.1 recruitment at this region, we hypothesized that *rs1247117* del would result in increased EIF3A expression. Accordingly, we found that EIF3A expression was elevated in *rs1247117* del cells relative to parental Nalm6 and SUPB15 cells, respectively (Fig. 6c, d, Supplementary Fig. 7b), further supporting an inverse relationship between PU.1 binding at *rs1247117* and EIF3A expression.

Because the risk G allele at *rs1247117* was also associated with vincristine resistance in primary ALL cells from patients, we additionally sought to determine the impact of the CRE deletion containing *rs1247117* on cellular response to vincristine treatment. We hypothesized that because the risk G allele is associated with enhanced PU.1 binding and resistance to vincristine, complete disruption of PU.1 binding in Nalm6 cells harboring the CRE deletion would show increased sensitivity to vincristine relative to parental Nalm6 cells. As predicted, Nalm6 cells with the CRE deletion exhibited significantly increased sensitivity to vincristine across a range of concentrations after 24, 48, and 72 hours of treatment (Fig. 6e–g), and we found consistent effects on cell viability in SUPB15 cells (Supplementary Fig. 7c). Consistent with enhanced sensitivity to vincristine, we also found increased caspase 3/7 activity in *rs1247117* del Nalm6 cells relative to parental Nalm6 cells after 72hrs and across a range of vincristine concentrations (Fig. 6h). These data suggest that a functional regulatory variant alters the binding affinity of a key transcription factor, PU.1, and disruption of this locus impacts EIF3A expression and vincristine sensitivity in ALL cells. To further validate our methodology utilizing CRISPR/Cas9 to delete CREs, we deleted CREs spanning two additional top variants, *rs7426865* and *rs12660691* (see Fig. 4e), that was associated with the ex vivo resistance to 6-mercaptopurine and dexamethasone, respectively, in primary ALL cells. Deletion of these CREs also impacted protein expression and sensitivity to the associated chemotherapeutic agent, thereby supporting our functional approach (Supplementary Figs. 8 and 9).

We next wanted to connect EIF3A directly to vincristine resistance. Given that EIF3A is an essential gene per the Broad Institute's DepMap, we opted to test the hypothesis EIF3A overexpression alone was sufficient to impact the Nalm6 cell response to vincristine. We, therefore, used lentiviral transduction to overexpress EIF3A in Nalm6 cells and compared EIF3A overexpression (EIF3A OE) cells to control infected cells (Nalm6 WT, Supplementary Fig. 10a). Using two independent infections of EIF3A OE, we found that at 48 hr and 72 hr, EIF3A OE cells were more sensitive to vincristine than Nalm6 WT cells (Supplementary Fig. 10b). These data suggest that EIF3A expression impacts the ALL cell response to vincristine, with higher expression sensitizing cells to the drug, and further establishes this gene as the likely target of the association.

## Discussion

Using MPRA, we systematically interrogated the functional effects of inherited noncoding variation associated with relapse, persistence of MRD after induction chemotherapy, and/or ex vivo chemotherapeutic drug resistance in childhood ALL. We refined our search to regulatory variants that were found in accessible chromatin sites in 161 ALL cell models, including primary ALL cells from patients, PDXs, and ALL cell lines, as those noncoding regions were likely to be participating in transcriptional regulation. Using MPRA we identified 556 functional regulatory variants showing reproducible and concordant changes in an allele-specific gene regulatory activity. To further explore the impact of these variants on gene regulation in ALL cell lines, we selected a subset of functional regulatory variants from MPRA that were within an accessible chromatin site in an ALL cell line. We overcame difficulties in associating promoter-distal functional regulatory variants with gene targets using promoter CHiC and found 19 variants with robust looping to a distal promoter, as well as 35 promoter-associated functional regulatory variants.

We identified *rs1247117* as the top functional regulatory variant, showing the highest geometric mean of differential transcription activity, which was identified in 9 of 10 ALL cell lines assayed by MPRA. We found that the allele present at *rs1247117* was a determinant of PU.1 transcription factor binding, with the risk G allele leading to greater chromatin accessibility and PU.1 binding affinity. Interestingly, the allele-specific activities as measured by MPRA and traditional dual-

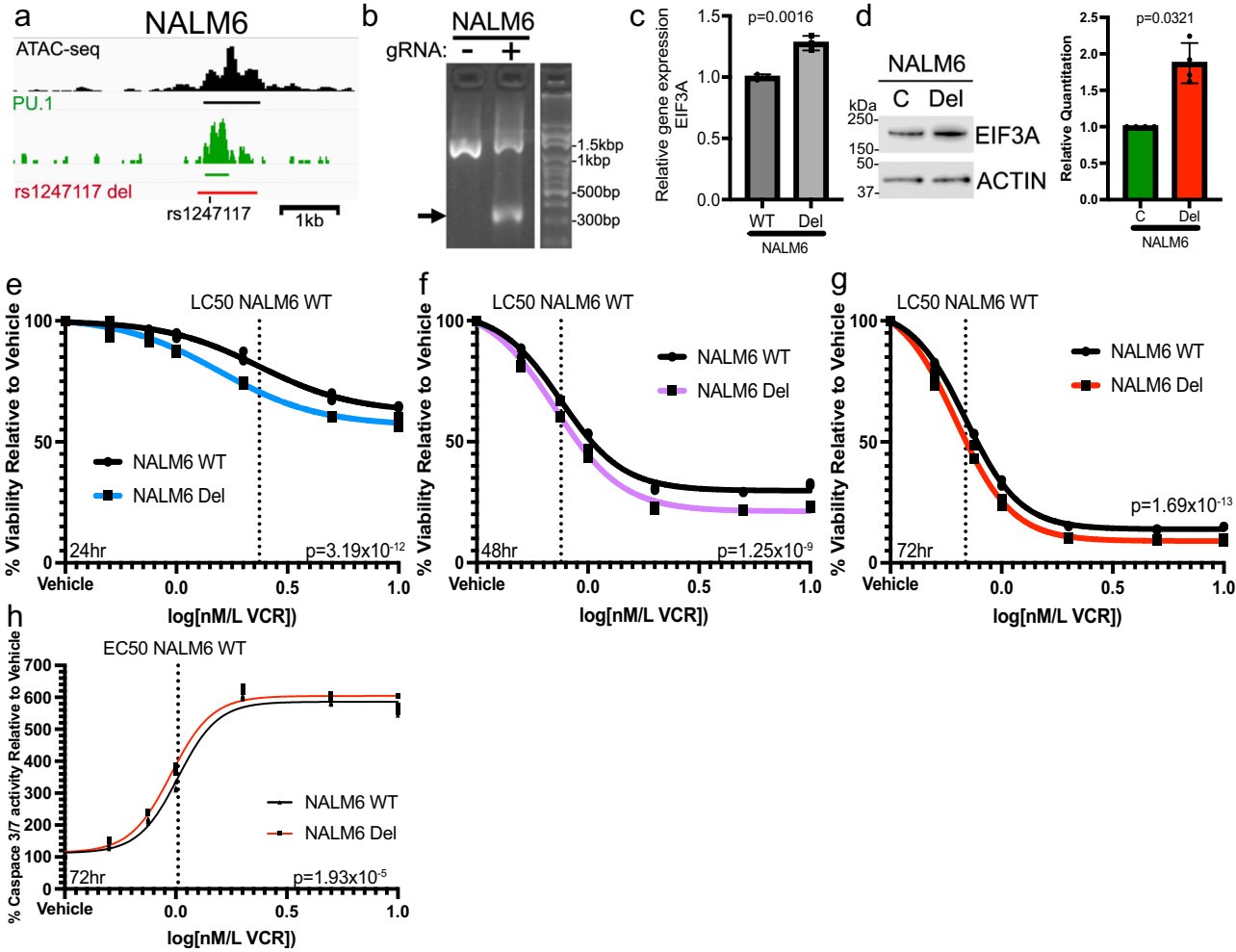

**Fig. 6 | Deletion of the CRE containing rs1247117 leads to increased EIF3A expression and sensitizes cells to vincristine. a** Diagram on the left showing the genomic context of the rs1247117 CRE deletion in Nalm6 cells in relation to chromatin accessibility, PU.1 binding and rs1247117. Black bar represents ATAC-seq peak, green par represents PU.1 peak, and red bar represents region deleted using CRISPR/Cas9 genome editing. **b** Gel shows validation of deletion using primers flanking deleted region. Arrow points to PCR fragment with deletion in heterogeneous Nalm6 cell pools harboring deletion compared to wild-type parental Nalm6 cells. **c** *EIF3A* gene expression is upregulated upon deletion of the CRE containing rs1247117. RT-qPCR data show the mean ± SD of three independent experiments. Two-tailed Student's *T* test. **d** Western blots and quantification showing increased EIF3A expression in rs1247117 del Nalm6 cells. Blots shown are representative of four independent experiments. Quantification data show the mean ± SD. Two-tailed Student's *T* tests, $n = 4$. **e–g** Drug sensitivity data comparing viability relative to vehicle treatment of wild-type parental Nalm6 cells and Nalm6 cells with rs1247117 CRE deletion after vincristine (VCR) treatment for 24 ($n = 3$), 48 ($n = 3$) and 72 ($n = 3$) hours at the indicated concentrations. Non-linear regression and $F$ test analysis indicate that these dose-response curves are significantly different. **h** Caspase 3/7 activity assays comparing Caspase activity relative to vehicle treatment of wild-type parental Nalm6 cells and Nalm6 cells with rs1247117 CRE deletion after vincristine (VCR) treatment for 72 hours at the indicated concentrations ($n = 3$). Dose-response curves of non-linear regression indicate that these curves are significantly different. Non-linear regression and $F$ test analysis indicate that these dose-response curves are significantly different. All source data are provided in the Source Data file.

luciferase reporter assays suggest that the reference G allele at *rs1247117* stimulates transcription more than the alternative A allele, and we suspect this is driven by greater PU.1 binding affinity in these episomal assays. However, our endogenous genetic manipulation that disrupted PU.1 binding at this locus in Nalm6 cells suggests that *EIF3A* expression is driven inversely to PU.1 binding. Moreover, complete disruption of PU.1 binding resulted in greater sensitivity to vincristine, which is consistent with the risk G allele contributing to both greater PU.1 binding affinity and vincristine resistance in primary ALL cells from patients. This discrepancy may be due to the ability of PU.1 to act in an activating or repressing manner on gene expression dependent on its genomic context and other transcriptional regulators present[66,67]. While not addressed within the scope of this work, our hypothesis is that MPRA and luciferase reporter assays, which are episomal and utilize a non-native minimal promoter, detected

transcription-activating PU.1 activities rather than the PU.1 repressive activities we detect within the endogenous locus. Collectively, these observations stress the importance of performing subsequent functional follow-up experimentation within an endogenous sequence context.

The *rs1247117* risk G allele frequency is ~20-30% in African (33%), American (29%), and East Asian (23%) populations, and only ~10% in Europeans (10%) and South Asians (11%). Notably, vincristine-associated neurotoxicity is more common in Europeans compared to African Americans[68]. However, this effect has been attributed to greater cytochrome P450 3A5 (*CYP3A5*) allele expression in African Americans compared to Europeans. Importantly, this variant was originally associated with the persistence of MRD after induction chemotherapy[3], and numerous reports demonstrate poorer outcomes in patients of African and American ancestries compared to White

populations[69–73]. Overall, our results are consistent with these studies and suggest that greater resistance to vincristine, which is typically given during induction chemotherapy, is a potential link between the risk G allele at *rs1247117* and the persistence of MRD in patients after induction chemotherapy.

Although the risk G allele at *rs1247117* is associated with decreased *EIF3A* expression and increased risk of MRD after induction chemotherapy and vincristine resistance, it remains unclear why *EIF3A* expression might impact vincristine efficacy. *EIF3A* expression has been correlated with cell cycle progression, and others have shown that *EIF3A* knockdown leads to decreased cell proliferation[74]. However, vincristine can act on microtubules to rapidly kill cells during G1 and later the mitotic spindle to arrest cells during metaphase, so increased *EIF3A* expression may facilitate cell cycle progression, thus increasing the rate at which metaphase mitotic spindles are disrupted by vincristine[75]. *EIF3A* expression has also been previously linked to chemotherapeutic sensitivity in both melanoma and lung cancer[76,77]. *EIF3A* expression led to decreased phosphorylation of ERK, supporting the effect of vemurafenib-induced MAP kinase signaling blockade, while *EIF3A* loss led to sustained activation of ERK and therapeutic resistance[76]. Interestingly, ERK activation is important in G1/S progression, and therefore, it follows that EIF3A-dependent inhibition of ERK may support the rapid killing of cells in the G1 phase shortly after initial vincristine treatment[75,78]. This notion is supported by significantly greater sensitivity to vincristine of ALL cells harboring greater *EIF3A* expression through disruption of a distal CRE after just 24 hours of treatment.

The regulatory variants assayed in this study were originally discovered from GWAS in patient samples, and most of our functional regulatory variant hits from MPRA were present in accessible chromatin sites found only in primary ALL cells from patients. Consequently, these data highlight both substantial differences in the chromatin landscape between immortalized cell lines and primary cells and a limitation of our study that relied on the functional exploration of top regulatory variants in ALL cell line models. An optimal approach would be to validate top functional regulatory variants in patient samples; however, this is not currently feasible due to the limited duration of patient sample viability in culture for genetic manipulation. Nonetheless, future implementation of promoter CHiC in patient samples can be used to map gene connectivity of promoter-distal functional regulatory variants found only in primary cells, and these gene targets can then be genetically disrupted in ALL cell line models for functional validation.

We acknowledge that our MPRA analysis only provides information on the allele-specific gene regulatory effects of these variants, and any connection with pharmacological effects would require additional validation, as we have provided and demonstrated with experimentation. Additionally, because genotyping was performed in primary ALL cells for variants associated with ex vivo drug resistance, we cannot exclude the possibility that some of our tested variants are somatic and not germline. Nonetheless, because we were comprehensive and provided extensive functional data across diverse ALL cell samples for this large set of variants, we believe that our rich dataset can additionally be utilized as an important resource for future GWAS studies on ALL pharmacogenomics. Namely, as future clinical trials are performed and additional GWAS variants impacting clinical phenotypes are discovered, our dataset can be queried to determine if newly discovered variants harbor functional effects on gene expression.

This translational work represents the largest functional investigation of inherited noncoding variation that is associated with diverse pharmacological traits in ALL to date. Our study identified hundreds of functional regulatory variants with significant, reproducible, and concordant allele-specific effects on gene expression, and further connected gene regulatory disruptions to differences in chemotherapy response through alterations in antileukemic drug sensitivity in ALL cells. Collectively, these data support the importance of noncoding, gene regulatory disruptions in the pharmacogenomics of ALL treatments. The further functional investigation of these regulatory variants and the discovery of additional inherited variants impacting therapeutic outcomes can be used by clinicians to tailor therapies based on a patient's unique genetic makeup through precision or personalized medicine.

## Methods

### Ethics

This study complies with all relevant ethical regulations, and the relevant study protocols were approved by the institutional review boards at St. Jude Children's Research Hospital and Children's Hospital of Philadelphia, as well as the Institutional Animal Care and Use Committee of St. Jude.

### Patient samples and consent

All patients or their legal guardians provided written informed consent. The use of these samples was approved by the institutional review board at St. Jude Children's Research Hospital. Ex vivo drug sensitivity assay and genotyping datasets were obtained from primary ALL cells of patients enrolled in St. Jude Children's Research Hospital (Memphis, Tennessee) Total Therapy XV (TOTXV, NCT00137111) and Total Therapy XVI (TOTXVI, NCT00549848) protocols[10,21]. RNA-seq data was obtained from primary ALL cells of patients enrolled on TOTXVI. Patient sample ATAC-seq data was obtained from published studies[55,56]. No associations between survival or treatment response and next-generation sequencing data were performed on samples from patients who are enrolled in ongoing clinical trials. No considerations for sex and/or gender were made for this study, and the pharmacogenomic traits were identified regardless of sex and or gender.

### Selection of SNVs impacting treatment outcome in patients from published GWAS

We chose 13 relapse SNVs with $p < 1 \times 10^{-5}$ from Yang et al.[4], 19 ancestry-specific SNVs relapse SNPs associated with relapse in both discovery and replication ALL patient cohorts ($p < 0.05$) from Karol et al.[5] and 3 SNVs associated with persistence of MRD with $p < 1 \times 10^{-6}$ from Yang et al.[3] ($n = 35$). In addition, we chose all SNVs ($n = 126$; Karol et al.[5] $n = 104$, Yang et al.[4] $n = 10$ and Yang et al.[3] $n = 12$) from these GWAS with nominal genome-wide association ($p < 0.05$) but that were additionally associated with ex vivo drug resistance ($p < 0.05$) in primary ALL cells from patients. Relapse SNVs additionally associated with ex vivo drug resistance were already identified by Karol et al.[5]. GWAS for ex vivo drug resistance ($p < 0.05$) using primary ALL cells from patients enrolled on TOTXV and TOTXVI protocols were performed on SNVs identified in Yang et al.[4] and Yang et al.[3] (see Methods below).

### Ex vivo drug sensitivity assays in primary ALL cells

Ex vivo drug sensitivity assay data was previously published[10,21]. The ex vivo drug sensitivity assays are previously described[21]. Briefly, primary leukemia cells were isolated from the bone marrow or peripheral blood of newly diagnosed ALL patients from St. Jude Total Therapy XVI protocol (TOTXVI, NCT00549848) and tested for antileukemic drug sensitivity by a 96-hour MTT assay using a range of drug concentrations. Primary ALL cells were treated with prednisolone ($n = 320$), dexamethasone ($n = 312$), bacterially derived L-asparaginase ($n = 335$), vincristine ($n = 323$), 6-mercaptopurine ($n = 344$) and 6-thioguanine ($n = 325$). Following drug treatment, the lethal concentration resulting in 50% viability (LC$_{50}$) was calculated for each patient sample.

## Genotyping in primary ALL cells

Genotyping data was previously published[21]. Briefly, genomic DNA was collected from primary ALL cells of patients enrolled on TOTXV and TOTXVI protocols using Blood and Cell Culture DNA kit (Qiagen), and SNP genotyping was performed using the Affymetrix Gene ChIP Human Mapping 500 K array or Affymetrix Genome-Wide Human SNP 6.0 array. Genotypes were called BRLMM algorithm in the Affymetrix GTYPE software (http://www.affymetrix.com/products/software/specific/gtype.affx) as previously described[79]. We excluded SNVs for call rates <95% among patients or minor allele frequencies <1%. SNPs were additionally filtered based on call rate and Hardy Weinberg equilibrium ($p > 1e-4$). Genome-wide genotypes were imputed using Topmed imputation server.

## Ex vivo drug resistance GWAS

Imputed genotypes with $r^2 \geq 0.6$ were tested for their association with LC50. To test the association between LC50 and genotype, LC50 were first log10 transformed and treated as a continuous dependent variable and genotype as independent variable. Genotypes were coded as 0, 1, and 2. The association between LC50 and genotype was tested using a general linear regression model for additive genotypic effect using PLINK. Two-tailed p-values were generated using a Wald test. Top 5 principal components were included as covariates in the regression model to control for population stratification. Ex vivo drug resistance GWAS was performed using primary ALL cells from patients enrolled on TOTXVI protocol. All statistical analysis was performed in R v4.0.2. SNVs that were nominally associated with ex vivo drug resistance ($p < 0.05$) and that were also identified as eQTLs in the same patient cohort ($p < 0.05$; see Methods below) were included in the MPRA analysis. For ex vivo drug resistance associations with treatment outcome SNVs (see Methods above), GWAS was performed from patients enrolled on TOTXV and TOTXVI protocols separately, and the GWAS results were then combined through meta-analysis using METAL.

## Cell lines, cell culture, and authentication

Leukemia cell lines (Nalm6 DSMZ ACC128, Jurkat DSMZ ACC282, B-ALL-1 DSMZ ACC742, 697 DSMZ ACC42, CEM DSMZ ACC240, REH DSMZ ACC22, P12-Ichikawa DSMZ ACC34, SEM DSMZ ACC546, Loucy DSMZ ACC394, DND41 DSMZ ACC525, HSB2 DSMZ ACC435, MOLT16 DSMZ ACC29, RS411 DSMZ ACC508, SUPB15 DSMZ ACC389) were cultured in RPMI 1640 + 10% fetal bovine serum. Cells were not cultured beyond 25 passages after we received them for experiments in this work. PCR-based mycoplasma testing determined that the cell lines were negative for mycoplasma contamination. STR profiling was confirmed before freezing down cell aliquots before passage 5 after we received them.

## eQTL mapping in primary ALL cells

RNA-seq data was obtained from a previous publication[21]. Briefly, total RNA from primary ALL cells was isolated using RNAeasy Mini kit (Qiagen), and mRNA sequencing using an Illumina HiSeq platform was performed by the Hartwell Center for Bioinformatics and Biotechnology at St. Jude Children's Research Hospital. Genotype information and RNA expression data from patients enrolled on TOTXVI protocol were correlated to identify eQTLs. QTL mapping was performed using multiple linear regression and log-transformed FPKM gene expression as the dependent variable and genotype as the independent variable. Genotypes were coded as 0, 1, and 2. Patient genetic ancestry was included in the linear model as covariates. Two-tailed p-values were generated using a Wald test. All statistical analysis was performed in R v4.0.2.

## Quantitative real-time PCR (qPCR)

See Supplemental Methods for more details.

## Dual-luciferase reporter assays

See Supplemental Methods for more details.

## MPRAs

MPRA oligo design, plasmid library construction, plasmid transfection, sequencing, and analysis are provided in Supplemental Methods.

## Epigenomic profiling in ALL cells

See Supplemental Methods for more details.

## PU.1 in vitro binding affinity assay

DNA pulldown assay was adopted from the previous article and performed using a modified protocol[80]. See Supplemental Methods for more details.

## CRISPR/Cas9 deletion

*rs1247117* deletions in Nalm6 were generated using CRISPR-Cas9 technology. See Supplemental Methods for more details.

## Nalm6 vincristine sensitivity assays

Drug viability assays were performed as previously with slight modification[22]. Additional information is provided in Supplemental Methods.

## Statistical analysis

See Supplemental Methods for more details on statistical analyses performed.

## Reporting summary

Further information on research design is available in the Nature Portfolio Reporting Summary linked to this article.

## Data availability

The ATAC-seq, promoter CHiC, HiChIP, RNA Pol II, and PU.1 genomic binding data generated in this study are publicly available in the GEO database under accession code GSE224204. The fresh patient sample ATAC-seq data used in this study are publicly available in the GEO database under accession code GSE226400. The frozen patient sample ATAC-seq data used in this study are publicly available in the GEO database under accession code GSE161501. H3K27Ac ChIP-seq data, "GSE175482 Nalm6 H3K27ac 0 hr merged.bw", used in this study are publicly available in the GEO database under accession code GSE175482. All other data generated in this study are provided in Supplementary Information and Source Data file. Source data are provided with this paper.

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

## Acknowledgements

We would like to thank the Hartwell Center at St. Jude for ATAC-seq, ChIP-seq, Cut & Run, promoter CHiC, HiChIP, and MPRA next-generation sequencing. We would also like to thank Jeremy Hunt and Brandon Smart for their technical support. This work was performed at St. Jude Children's Research Hospital and was also conducted in part by the ECOG-ACRIN Cancer Research Group. This work is supported by the National Cancer Institute (R01CA234490—Daniel Savic, P30CA021765—St. Jude Children's Research Hospital, U10CA180820—ECOG-ACRIN Medical Research Foundation, UG1CA232760—MAYO Clinic, Rochester and UG1CA189859—Montefiore Medical Center), the National Institute of General Medical Studies (P50GM115279—St. Jude Children's Research Hospital) and the American Lebanese Syrian Associated Charities (ALSAC). The content is solely the responsibility of the authors and does not necessarily represent the official views of the National Institutes of Health.

## Author contributions

Kashi R.B., R.J.M.: investigation, writing—original draft, review, and editing, visualization, supervision, methodology, conceptualization, Kelly R.B.: formal analysis, data curation, D.C.F., B.S.H., J.D.D., B.P.B., S.Y.: investigation, W.Y., C.C.: formal analysis, K.R.C., C.S.M., E.J., S.M.K., W.S., H.I., S.J., C.P., J.J.Y.: resources, E.P., M.R.L., M.V.R., W.E.E.: resources, funding acquisition, S.P.-M.: methodology, D.S.: conceptualization, methodology, supervision, project administration, funding acquisition, writing—original draft, review, and editing.

## Competing interests

The authors declare no competing interests.

## Additional information

[1]Hematological Malignancies Program, St. Jude Children's Research Hospital, Memphis, TN 38105, USA. [2]Department of Pharmacy and Pharmaceutical Sciences, St. Jude Children's Research Hospital, Memphis, TN 38105, USA. [3]Center for Advanced Genome Engineering, St. Jude Children's Research Hospital, Memphis, TN 38105, USA. [4]Department of Cell and Molecular Biology, St. Jude Children's Research Hospital, Memphis, TN 38105, USA. [5]Graduate School of Biomedical Sciences, St. Jude Children's Research Hospital, Memphis, TN 38105, USA. [6]Department of Advanced Pediatric Medicine, Tohoku University School of Medicine, Tokyo, Japan. [7]Alliance Hematologic Malignancy Biorepository; Clara D. Bloomfield Center for Leukemia Outcomes Research, Columbus, OH 43210, USA. [8]Department of Leukemia, The University of Texas MD Anderson Cancer Center, Houston, TX, USA. [9]Albert Einstein College of Medicine, New York, NY, USA. [10]Division of Hematology, Department of Medicine, Mayo Clinic, Rochester, MN 55905, USA. [11]Comprehensive Cancer Center, University of Chicago Medicine, Chicago, IL, USA. [12]Department of Oncology, St. Jude Children's Research Hospital, Memphis, TN 38105, USA. [13]Department of Biostatistics, St. Jude Children's Research Hospital, Memphis, TN 38105, USA. [14]Integrated Biomedical Sciences Program, University of Tennessee Health Science Center, Memphis, TN 38163, USA. [15]These authors contributed equally: Kashi Raj Bhattarai, Robert J. Mobley. ✉e-mail: daniel.savic@stjude.org

