## [Peer Review File · Nature Communications]

Investigation of inherited noncoding genetic variation impacting the pharmacogenomics of childhood acute lymphoblastic leukemia treatmentREVIEWER COMMENTS

Reviewer #1 (Remarks to the Author): expertise in paediatric ALL genomics

The paper entitled "Investigation of inherited noncoding genetic variations impacting the pharmacogenomics of childhood lymphoblastic leukemia treatment" aims to investigate the gene regulatory potential of inherited noncoding variants associated with chemotherapy resistance and the outcome to ALL. The authors identified rs1247117 (mapping to a distal intergenic region near the CACUL1 gene) as the top functional regulatory variant. Notably, deletion of the cis-regulatory element (CRE) containing rs1247117 correlated with up-regulated E1F3A expression. The authors demonstrated that a CRE deletion involving this SNV correlated with increased sensitivity to vincristine.

This paper add valuable data to the knowledge of the molecular mechanisms underlying the chemotherapy resistance in ALL.

Concerns,

Line 179-188: Base on those findings described in lines 179-183, I could not see the rational "... these data suggest that the chemotherapeutic drug sensitivity and patient treatment outcome SNVs tested were heavily enriched for functional regulatory variants with the potential to impact gene regulation". Did the authors find a profile associated with resistance to specific drugs? What SNVs were shared among the group of cell lines that are resistant to specific antileukemic drugs?

Line 200: "... chromatin in an ALL cell line..." Which cell line was used? Is this cell line positive to specific fusion genes or resistant to a specific anti-leukemic treatment?

Line 227-230. Is this phenomenon a potential explanation of the association among well known SNVs (ie. ARID5B rs10821936, rs10994982) in ALL and treatment resistance?

Line 237-239. The rs1247117 was selected bases on the LD values with, rs1312895 and rs1247118, but I could not find published data regarding the association among those SNV with ALL, persistence of MRD, or drug resistance. Please, explain it.

Line 241. Please state whether rs1247117 binds to the EIF3A promoter only in Nalm6 B-ALL but no other ALL cell line or the rationale of Nalm6 B-ALL as study cell model. Do this results could be extrapolated to ALL cases harboring rs1247117 G allele?

Adding a table enlisting the most significant SNVs including their closest genes could be of the interest of the readers.

Additionally, I am wonder if the authors did no find interesting results regarding variantes located in ARID5B, NUDT15, etc, which are frequently associated with ALL.

Reviewer #2 (Remarks to the Author): expertise in bioinformatics and non-coding variants

This study investigates susceptability loci for ALL using high-throughput functional genomics. This is an interesting example of an MPRA-based approach integrating general data with ALL-specific data to arrive at a reduced set of loci for detailed follow-up.

Results, line 132: why is the number of patients given as a range (312-244)?

Line 134: Source of eQTLs from primary ALL cells?

Line 198-199: "further functional investigation of variants in primary ALL cells is currently intractable" - is this because MPRA cannot be carried out in primary cells because of technical problems around propagation/transfection?

Line 210: Do the "over 500 SNVs" correspond to the 556 SNVs with "significant and concordant

allele-specific activities in at least 3 ALL cell lines" (line 193-4)? If so probably better to stick with the 556 figure for clarity.

The "G" allele at rs1247117 is quite common with some variability between populations - is there anything known about response to vincristine (e.g. neurotoxicity) that may vary according to genetic population structure that could corroborate these results?

Figure 5h, also Supplementary Methods lines 317-320: should the repeated T-tests here be controlled for false discovery rate via multiple testing correction?

I understand that the aim of this paper was to dissect one particular "hit" (rs1247117) in detail, but it would be interesting to hear more about the other hits. Is there information here about themes underlying genetic susceptibility to ALL?

Reviewer #3 (Remarks to the Author): expertise in epigenetics methods and HiC

The goal of the study by Bhattara et al. was to test the premise that the underlying non-coding genetic variations and gene regulatory factors would impact the diverse pharmacological traits found in ALL. Based on prior information and GWAS analysis, SNVs were identified and selected for this study. Therefore, the authors tested their interrogation of GWAS regulatory variants that map accessible chromatin sites in ALL by ATAC seq analysis and use MPRA to narrow down the scope of non-coding variants to test. Some of which are believed to be associated with ex vivo chemotherapeutic drug resistance in primary ALL cells from patients and/or ALL treatment outcome (i.e., relapse and persistence of MRD) from patient info. The authors then attempt to correlate these results with promoter ChIP to identify candidate target genes of functional regulatory variants with significant allele-specific effects on reporter gene expression. Lastly, the authors then investigate the impact of the top regulatory variant on transcription factor binding, neighboring gene expression and chemotherapeutic drug resistance in ALL cell line cultures. Regarding the impact, the authors claim that this study represents the largest functional investigation of regulatory variants impacting the pharmacogenomics of chemotherapy treatment and fills an unmet need for large-scale functional examinations of regulatory GWAS variants associated with pharmacological resistance. There are some modest doubts about that claim.

Generally noted on the positive side of the study, the flow of the manuscript appears logical and straightforward to follow. Moreover, the authors make the reader aware of the discrepancies found between the screening results and functional validation of the data provided. The authors do provide some moderate insight from the functional screen of the GWAS SNVs lists and appear fairly comprehensive, with some limitations noted. Generally, the biochemical studies appear somewhat robust with some questions noted below. The authors make efforts to confirm the PU1 immunoprecipitation studies, chromatin IPs, and biotin-DNA pull down, where some effort went into confirming the interaction of PU.1 and its association with the SNV region. In the sense that the authors have been cautious in interpreting their results. Nonetheless, a marginal issue is with the nature of the sequence depth obtained and data analyzed, why not evaluate some additional SNVs to affirm the strength and statistical power of MPRA screening? I believe this maybe be advantageous to stress the strength of the authors' approach.

Despite the strengths noted, some clarity is definitely required to make some sense of the author's datasets used. For instance, all the patient studies (drug-testing, DNA-GWAS, RNA, ATAC) was done by their core. Moreover, data was not released or available for this manuscript review (at least this reviewer couldn't retrieve or identify this information). This is exemplified in Figure 1A, where is unclear the authors' relationship to this dataset a, and I do not see any information about this aspect aside from some sample numbers provided. Additionally, many of the methods used were vaguely and not detailed sufficiently to really understand the manner in which data might give variation and discrepancy in their results This maybe remedied merely by providing additional supplementary information and data to support the authors' description of their functional assays. For instance, what were the quality control (QC) metrics even basic QC metrics were not clear since the authors are mentioning discrepancy in their results. Such as their reporter assay (MPRA and luciferase assays) and their functional validation by CRISPR for instance. I believe readers

would like to be aware how and why discrepancies in their data may have arisen. Of course, use of reporter assays, which does not reflect the margins of a promoter used or its native context could be one reason (for instance, where a CRISPR deletion cannot necessarily equate to an actual SNVs, see below detail), but if the expected result is in the opposite direction, It questions how confident readers would be for the authors to apply the MPRA as a screening strategy.

Specific Comments:

Figure 3C and within the discussion the authors mentioned ATAC seq results from patients and primary cells. This appears to differ significantly between them. One wonders if that's due to a quality issue between cell lines and primary cells. If the authors think differences arise from cell line vs primary cell, then fresh/frozen primary ALL should be very similar. The authors should perform QC and define their correlation. As mentioned above authors should provide some QC data and their metrics to show all the ATAC result (TSS enrichment and FRIP scores, etc.) are at comparable levels and their stats are reasonable, otherwise, direct normalized counts comparison in Fig S4 would not be meaningful whereas the variable regression of the data would be needed. This reader noticed that they are using FAST-ATAC for all of their fresh/frozen primary and cell lines. This appears as an OK method for cryopreserved frozen samples, but for fresh samples, especially cell lines, this doesn't seem suitable. Lacking the cell lysis step for fresh healthy cell line cells may result in permeabilization and therefore a lack of proper Tn5 tagmentation, which essentially could explain lack of their results confirming the cell line association with the SNVs.

Figure 2: A question to consider is whether the authors could do primary cell ALL MPRA experiments, the transfection and RNA harvest are done within 24h. Understandably, use of ALL primary cells maybe challenging to perform MPRA, but it has been performed with other primary cancer lines and certainly primary lymphoblasts. It would be very attractive to readers to appreciate how ALL primary cell and cell lines compare, as illustrated in Figure 2D. If the cell line is the culprit for some of this discrepancy, then it would be definitely encouraging to see that some primary cell or patient-derived cell MPRA data exhibiting the same alt/ref expression changes. Expanding MPRA to a broader sample type (not just cell line) would be a nice leap forward for other disease-type screening.

Figure 3b, It is not clear to this reader where the non-cell line data is coming from, (??). In the methods section, MPRA was described as only being tested in ten cell lines. Please provide this information. In Figure 3c, since the authors may want to show the shared SNVs in open chromatin between different sample types. Maybe authors should consider an upset plot?

Figure 4 a-c, It maybe beneficial to see how the non-promoter-associated SNV performed in panel "b" as well, since this is the largest group. In panel "b", the log2 difference from 2 groups does not look drastically different, even the p value challenges significance. It is worth some explanation for the reader to understand better.

Figure 4 e, it would increase impact and the strength of the approach if authors could test a few additional top SNVs from their functional assay. Since the authors already have the HiC information this may further strengthen their data and approach.

Figure 5 a: This maybe an issue for presentation but the IGV tracks should provide a normalized count read across the tracks for each of the tracks shown. It is hard to appreciate the values and levels of enrichment. For instance are they [0-10], [0-100], etc.. It would be important to have this available for readers

Figure 5 e-g: One question. Why did the authors delete such a large genomic regions for the functional assay? Couldn't they perform a CRISPR-knockin to introduce the SNV? Such would introduce significant artifacts for chromatin structure, chromatin accessibility, and TF binding. These findings are likely very different by deleting almost 1kb region. Reasoning is not clear to this reader. Moreover, this may also explain why this functional validation does not match their own reporter assays.

Figure 5h, The authors need to show the zero (0) -time point to show the same viability to start off with, they also need a DMSO control (or just vehicle) group for both WT and DEL samples just to provide base line viability for both. Hence the reasoning is whether the authors truly statistically account for the significance difference between groups where it appears small for such significant **** $p < 0.0001$ values noted from $n=3$. Please clarify.

Reviewer #4 (Remarks to the Author): expertise in HiC analysis

NCOMMS-23-06280A

The manuscript by Bhattarai, Mobley et al. provides a functional investigation of noncoding variants identified in GWAS studies and ex-vivo drug resistance assays. The approach to validate these putative CRE elements was to perform an MPRA assay on 1696 SNVs and evaluate which of them altered reporter expression. To further prioritize SNVs they combined their results with chromatin accessibility data from primary samples and cell lines, with the assumption that functional elements should be characterized by open chromatin. To identify potential targets of these putative CREs they performed promoter Capture-C and checked which of them were associated with analyzed promoters. Finally, they focused on a specific SNV (rs1247117) and performed molecular and biochemical assay to evaluate its role on transcriptional regulation and vincristine chemoresistance. Despite this approach being interesting and innovative, there are major concerns with some aspects of this work, which may hamper the publication on a high impact journal as Nature Communications. The authors should thus address the following points:

- In line 186 the authors wrote "Overall, these data suggest that the chemotherapeutic drug sensitivity and patient treatment outcome SNVs tested were heavily enriched for functional regulatory variants with the potential to impact gene regulation". This conclusion is too strong based on the data presented:
 - o MPRA experiment was performed using cell lines, while most detected variants were identified from primary patients. The functional relevance of these elements could thus be biased by the homogeneity of the cell lines populations, for instance in terms of transcription factors expression. To claim that the experiment enriched for disease relevant variants it would necessary to perform the assay on primary ALL derived cultures, which better express intra-tumoral heterogeneity.
 - o The robustness of the results across the cell lines tested is not so evident from the figures. In fig.2c B-ALL lines manifest a high degree of variability regarding the number of SNVs with differential activity. How do the authors explain that? T-ALL lines seem more uniform. Does this reflect differences in terms of transcription factors expression? Did the authors perform a motif analysis of the candidate sequences used for the assay?
 - o In fig.2d and suppl.1, correlation plots are centered with higher density around (0;0), suggesting that most pairwise comparison occurred with a log₂FC of 0 in both cell lines A and B. This suggests that statistical significance is mainly driven by concordant minimal changes in activity between Alt/Ref and does not really support the idea that these SNVs had a biological relevance.
 - o The authors should provide the RNA counts for each SNVs to assess the relevance of the log₂ fold changes.
 - o It would thus be better to modify the conclusion from line 186 and present the data just as the workflow that allowed the prioritization of the variants that were further investigated
- The frequency of concordance across the cell lines should be clarified, plotting how many SNVs were concordant in 1, 2, 3, .. 10 cell lines in a frequency distribution. It would help to strengthen the conclusions.
- It should be better specified in the text that the luciferase assay was performed on only 11 variants. Claiming that these results validate the MPRA experiment and its robustness is somewhat excessive. Also reporting that "over 500 SNVs had reproducible and concordant effects" is excessive, considering that some/many (it is not clear from the data presented) of the variants may be concordant only in 1/3 of the lines tested.
- The results presented in fig.4a provide evidence that most of the variants that were prioritized were not nearby a promoter or connected to a distal one. These data highlight an evident lack of discovery power of the promoter Capture-C approach. It would be important to either analyze publicly available or perform Hi-C or H3K27ac HiChIP experiment to collect more CRE interaction data. It is possible that many of the 136 "non-promoter associated" variants also not associated to distal promoters are associated with other enhancers, directly connected to a promoter. It would be important to address this point, because if these 136 elements actually do not result connected

even to other enhancers, they may represent false positive hits of the MPRA assay. In that case, this information would be necessary to evaluate the relevance of this approach.

- The in vitro assay presented in fig.5d supports the idea the A/G variant has reduced binding affinity for PU.1. However, its functional impact in the endogenous chromatin context is not addressed. It would be important to introduce the variant in a wild type cell line and perform PU.1 ChIP experiments to evaluate changes in PU.1 binding. Alternatively, Nalm6 cells could be genetically modified to harbor two A/A alleles and reproduce the primary data presented in fig. suppl 4c, performing both PU.1 ChIP and ATAC-seq. These in vitro models could also be used to validate the result of fig. 5f-g-h.

- The experiments presented in fig.5f-g-h are not sufficient to sustain the biological relevance of the variant analyzed, for key reasons:

- o RNA changes evaluated by qPCR are relatively small, despite being statistically significant. The authors should also provide evidence at protein level for the two genes.

- o The changes in relative viability of cells treated with VCR are relatively small, despite being statistically significant. The authors should also perform other independent assay to evaluate changes in cell proliferation or viability.

- o It would be important to perform similar experiments in cells with ectopic inducible augmented expression of CACUL1 and EIF3A to demonstrate their relevance in VCR resistance.

- rs1247117 was suggested to have an impact on MPRA assay in multiple cell lines, but the authors performed validation experiments only on Nalm6. They should extend their findings at least to another cell line to strengthen their conclusions.

Minor comments:

- The authors should provide a clearer list of samples used for analysis in a supplementary table, defining how many B and T-ALLs

- Suppl.table 2: it is not clear which are the 35 positive control SNVs and the 31 that had a significant allelic effect as indicated in the text

- Fig.3c: colors used for boxes of "Cell Line" and "PDX+Patient sample+Cell line" are difficult to distinguish

Reviewer #5 (Remarks to the Author): expertise in GWAS methodology in ALL

This paper represents a substantial amount of work to define a set of inherited genetic variants that are associated with response to treatment in ALL patients. This effort has the potential for high clinical significance in the screening of ALL patients for different treatments. Further, this paper combines an innovative gene-mapping strategy using ex vivo drug resistance assays in ALL cells with downstream functional analyses to identify regulatory variants for treatment response.

Since all of the downstream analyses are dependent on defining a set of inherited genetic variants, my review will primarily focus on this stage of the paper.

1) Genotyping for the treatment response GWAS was done using DNA obtained primary ALL cells. However, since they are using cancer cells there is some concern that some mutations will be somatic. These are primarily expected to be large indels/CNVs, but there may be some point mutations as well. This caveat of somatic mutations should either be covered in the Discussion or methods used to exclude these should be included in the Methods.

2) No information as to the number of variants included in the treatment response GWAS or the specific quality control steps applied to the genotype data were included in the Methods except for call rate and MAF.

3) It appears as if all SNVs with $p < 0.05$ were included from the treatment GWAS, but this threshold seems too liberal even with extensive functional downstream analyses. By combing data

from Supplemental Files 2 and 3, it appears as if the 53 SNVs that have reproducible transcriptional effects also have highly significant associations with one or more treatment response it is not clear why such a liberal p-value threshold was used. It would be helpful to present these association statistics for the 53 variants in one table. It would be interesting to see how many of these 53 variants would have been selected if a more stringent p-value threshold was employed.

4) Figure 1 is extremely helpful to understand the overall study design for the selection of SNVs. However, it would be helpful to have the number of SNVs that were contributed from each source (Figure 1b).

RESPONSE TO REVIEWERS' COMMENTS

Reviewer #1 (Remarks to the Author):

The paper entitled “Investigation of inherited noncoding genetic variations impacting the pharmacogenomics of childhood lymphoblastic leukemia treatment” aims to investigate the gene regulatory potential of inherited noncoding variants associated with chemotherapy resistance and the outcome to ALL. The authors identified rs1247117 (mapping to a distal intergenic region near the CACUL1 gene) as the top functional regulatory variant. Notably, deletion of the cis-regulatory element (CRE) containing rs1247117 correlated with up-regulated E1F3A expression. The authors demonstrated that a CRE deletion involving this SNV correlated with increased sensitivity to vincristine.

This paper add valuable data to the knowledge of the molecular mechanisms underlying the chemotherapy resistance in ALL.

Concerns,

1. Line 179-188: Base on those findings described in lines 179-183, I could not see the rational “... these data suggest that the chemotherapeutic drug sensitivity and patient treatment outcome SNVs tested were heavily enriched for functional regulatory variants with the potential to impact gene regulation”. Did the authors find a profile associated with resistance to specific drugs? What SNVs were shared among the group of cell lines that are resistant to specific antileukemic drugs?

We apologize for the confusion. The MPRA is disconnected from any pharmacological phenotype or effect, and solely assesses transcriptional output (i.e., effects on gene expression) and was therefore solely used to identify differences in transcriptional output between two alleles at SNVs. No specific enrichment of a gene profile is associated with the data. We have now revised this statement in the manuscript. We do now also perform gene associations for all 556 reproducible and concordant variants using GREAT (McLean et al. *Nat Biotech* 2010; PMID: 20436461) in **Supplementary Table 4**, as well as list drug phenotypes for each variant. We have also performed pathway analysis using GREAT on all 556 variants in **Supplementary Table 5**.

2. Line 200: “... chromatin in an ALL cell line...” Which cell line was used? Is this cell line positive to specific fusion genes or resistant to a specific anti-leukemic treatment?

We apologize for the lack of clarity in this statement. We intended to declare that we were focusing on variants that could be found within accessible chromatin (i.e., ATAC-seq sites) in one of the 14 ALL cell lines for which we had collected ATAC-seq data. This is important as further experimentation, such as functional validation and/or genetic alterations (CRISPR genome editing, etc.), are largely intractable in primary ALL cells or PDXs *ex vivo* and can therefore only be performed in ALL cell lines. Consequently, we had to ensure that an accessible chromatin site was present in a cell line for downstream functional validation. We have edited the manuscript to better explain this point.

3. Line 227-230. Is this phenomenon a potential explanation of the association among well known SNVs (ie. ARID5B rs10821936, rs10994982) in ALL and treatment resistance?

GWAS hits predominantly localize to non-coding sequence space, and many causal variants are therefore believed to be linked to their phenotypic effects via gene regulatory defects. We have discussed these ideas in a past review article (Sakabe et al. *Genome Biology* 2012, PMID: 22269347). Notably, we have also recently identified gene regulatory defects for GWAS variants for ALL disease risk at the *ARID5B* gene locus (Zhao et al. *JNCI* 2022, PMID: 35575404). Collectively, we believe that gene

regulatory effects are a potential explanation for many, if not most, non-coding GWAS associations, and our current findings are consistent with this notion.

4. Line 237-239. The rs1247117 was selected based on the LD values with, rs1312895 and rs1247118, but I could not find published data regarding the association among those SNVs with ALL, persistence of MRD, or drug resistance. Please, explain it.

The initial association with persistence of minimal residual disease (MRD) was identified in Yang et al. *JAMA* 2009 (PMID:19176441). These two variants are located in supplemental table 1 of Yang et al. *JAMA* 2009 among a list of top SNVs associated with end-of-induction MRD. These variants were also associated with resistance to vincristine using *ex vivo* drug sensitivity data from primary ALL cells of patients enrolled on St. Jude Total Therapy XVI and XV protocols (see revised **Methods** for criteria for variant inclusion).

5. Line 241. Please state whether rs1247117 binds to the *EIF3A* promoter only in Nalm6 B-ALL but no other ALL cell line or the rationale of Nalm6 B-ALL as study cell model. Do these results could be extrapolated to ALL cases harboring rs1247117 G allele?

There is statistically significant evidence for rs1247117 contact with the *EIF3A* promoter in Nalm6 cells. Because of this fact, and because rs1247117 resides in an accessible chromatin site in Nalm6 cells, we used Nalm6 cells as the cell model to functionally investigate this variant further. Of course, functional evaluations in primary cells would be ideal. However, genetic manipulations in primary ALL cells are currently inefficient and *ex vivo* experimentation is largely intractable. We therefore had to identify suitable ALL cell line models for further functional interrogation. Our investigation in Nalm6 suggests that the G risk allele more robustly recruits PU.1, which negatively impacts the expression of *EIF3A*. In the revised manuscript, we further demonstrate that the protein expression of *EIF3A* is impacted by both deletion of the ATAC-seq site spanning rs1247117 and by CRISPR-induced allele swapping in Nalm6 cells (see **reply to Reviewer #4, Comment #9** below). Moreover, we further demonstrate *EIF3A* as the target gene of the association, with *EIF3A* over-expression leading to greater vincristine sensitivity (see **reply to Reviewer #4, Comment #12** below).

Although we identify an interaction with the *EIF3A* promoter in Nalm6 cells only, our data does not exclude the possibility that rs1247117 contacts the *EIF3A* promoter in other ALL cells. False negative results are common with functional genomic assays, as no single assay can identify all active regulatory elements or 3D interactions present in a cell. Promoter capture Hi-C is therefore likely to have missed many true interactions (i.e., because of limited power, technical artifacts, etc.), including interactions with the *EIF3A* promoter in other cell models. Accordingly, in the revised manuscript, we validate the effects of rs1247117 on vincristine resistance and *EIF3A* expression in SUPB15 cells even though promoter capture Hi-C failed to identify an interaction in this cell line (see **reply to Reviewer 4, Comment #13** below). Collectively, we expect that this mechanism would be true in other ALL cell models, including primary ALL cells from patient samples harboring the risk G allele.

6. Adding a table enlisting the most significant SNVs including their closest genes could be of the interest of the readers.

We thank the Reviewer for this suggestion. We have now added gene associations with the 556 reproducible and concordant variants we identified using GREAT (McLean et al. *Nat Biotech* 2010; PMID: 20436461) and also included promoter capture interactions (see our **reply to Reviewer #4, Comment #8** for H3K27ac Hi-ChIP analysis and **Supplementary Table 4**).

7. Additionally, I am wonder if the authors did no find interesting results regarding variantes located in ARID5B, NUDT15, etc, which are frequently associated with ALL.

We included 35 control SNVs associated with ALL disease risk (e.g., ARID5B variants), and 31 out of 35 (89%) of these variants were identified as having significant allele-specific effects in at least one ALL cell line (see **Supplementary Table 2**). In addition, 10 of the 35 control ALL risk variants (29%) were identified as having reproducible effects in ≥ 3 ALL cells line and consistent allele-specific effects (see **Supplementary Table 4**). Our results are also in line with the results from independent investigations regarding the allele-specific effects of these ALL disease risk variants (e.g., rs3824662 at GATA3 locus; Yang et al. *Nature Genetics* 2022, PMID: 35115686). However, outside of using these variants as controls and the descriptions provided above, we did not focus on reporting additional effects or performing additional analyses on these variants as this was beyond the scope of our current work which focused on functionally interrogating associations relevant to therapeutic outcomes. We include the MPRA results of these positive control variants in **Supplementary Tables 2** and **4** for groups that want to pursue additional investigations of these variants.

Reviewer #2 (Remarks to the Author):

This study investigates susceptibility loci for ALL using high-throughput functional genomics. This is an interesting example of an MPRA-based approach integrating general data with ALL-specific data to arrive at a reduced set of loci for detailed follow-up.

1. Results, line 132: why is the number of patients given as a range (312-244)?

For the *ex vivo* drug sensitivity assays, there were differences in the total number of ALL patient biospecimens (i.e., primary ALL cells) used for each antileukemic agent (see revised **Methods**). The differences in biospecimens used largely stem from limited availability of biospecimens for the assessment of antileukemic drug sensitivity for all agents available and/or from technical failures of the *ex vivo* drug sensitivity assay.

2. Line 134: Source of eQTLs from primary ALL cells?

These came from RNA-seq and genotyping array results generated in primary ALL cell biospecimens from patients enrolled on the St. Jude Total XVI protocol. This is the same patient cohort wherein *ex vivo* drug sensitivity studies on patient bio-specimens were performed. We have included this information in the revised **Methods** section.

3. Line 198-199: "further functional investigation of variants in primary ALL cells is currently intractable" - is this because MPRA cannot be carried out in primary cells because of technical problems around propagation/transfection?

We were largely referring to CRISPR-based genetic manipulations and deletions for downstream functional validation studies because, as the Reviewer correctly pointed out, primary ALL cells do not propagate in culture. Moreover, transfection is problematic in lymphoblastoid cells, and especially in primary ALL cells wherein cell viability is also limiting *ex vivo*.

Despite these challenges, we have now performed MPRA *ex vivo* using patient-derived xenografts of primary ALL cells (ALL PDXs) from two patients, albeit the data are not as robust as the ALL cell line data because of the technical problems described above (see **reply to Reviewer #3, Comment #2** and **Figure 3 below**, as well as **Supplementary Figure 2b-c** and **Supplementary Table 3** of the revised manuscript).

4. Line 210: Do the "over 500 SNVs" correspond to the 556 SNVs with "significant and concordant allele-specific activities in at least 3 ALL cell lines" (line 193-4)? If so probably better to stick with the 556 figure for clarity.

The Reviewer is correct, and we have now edited the manuscript to state 556.

5. The "G" allele at rs1247117 is quite common with some variability between populations - is there anything known about response to vincristine (e.g. neurotoxicity) that may vary according to genetic population structure that could corroborate these results?

This is a great point made by the Reviewer. Indeed, the rs1247117G allele frequency is ~20-30% in African (33%), American (29%) and East Asian (23%) populations, and only ~10% in Europeans (10%) and South Asians (11%). Notably, vincristine-associated neurotoxicity is more common in Europeans (34.8%) compared to African Americans (4.8%; Renbarger et al. *Pediatric Blood & Cancer* 2008, PMID: 18085684). However, this effect has been attributed to greater cytochrome P450 3A5 (CYP3A5) allele expression in African Americans compared to Europeans. Importantly, this variant was originally

associated with persistence of minimal residual disease (MRD; Yang et al. *JAMA* 2009, PMID:19176441), and numerous reports demonstrate poorer outcomes in patients of African and American ancestries compared to White populations (Yang et al. *Nature Genetics* 2011, PMID: 21297632; Kadan-Lottick et al. *JAMA* 2003, PMID: 14559954; Bhatia et al. *Blood* 2002, PMID: 12200352; Pollock et al. *JCO* 2000, PMID: 10673523; Pui et al. *JAMA* 2003, PMID: 14559953). Our results are consistent with these independent observations on differences in treatment outcome among populations. Moreover, our data suggest that greater resistance to vincristine, which is typically given during induction chemotherapy, is a potential link between the risk G allele and persistence of MRD in patients after induction chemotherapy. We have included these data on disparities in treatment outcome among human populations in the revised manuscript.

6. Figure 5h, also Supplementary Methods lines 317-320: should the repeated T-tests here be controlled for false discovery rate via multiple testing correction?

We apologize for the confusing display and analysis of these data. We have re-analysed the data using dose-response curves with non-linear regression and identified significant differences between the curves of deletion and parental cells ($p < 0.0001$ for the two curves at each timepoint; see **Figure 1** below and **Figures 6e-g** of the revised manuscript).

7. I understand that the aim of this paper was to dissect one particular "hit" (rs1247117) in detail, but it would be interesting to hear more about the other hits.

We have now performed functional investigations of two additional top variant hits, rs7426865 and rs12660691. rs7426865 was associated with resistance to 6-mercaptopurine. We deleted the accessible chromatin site spanning rs7426865 in SUPB15 cells in heterogeneous cell pools. The deletion resulted in decreased expression of the proteins encoded by several connected and nearby genes and impacted sensitivity to 6-MP (see **Supplementary Figure 8** of revised manuscript). rs12660691 was associated with resistance to dexamethasone. We also deleted the accessible chromatin site spanning rs12660691 in SUPB15 cells in heterogeneous cell pools. rs12660691 deletion resulted in decreased ARHGAP18 expression and impacted sensitivity to dexamethasone (see **Supplementary Figure 9** of revised manuscript). Collectively, these validations support the role of these loci in antileukemic drug resistance.

8. Is there information here about themes underlying genetic susceptibility to ALL?

Overall, our data demonstrate allele-specific effects for many of these variants (31 of 35, 89%; 29% with reproducible and consistent effects in >2 ALL cell lines) and therefore suggest that gene regulatory defects are likely to play a role in ALL disease susceptibility. We primarily used these variants as controls since they have been previously investigated, and we note where our data is concordant with previous investigations (e.g., rs3824662 at *GATA3* locus; Yang et al. *Nature Genetics* 2022, PMID: 35115686). Because of our focus on pharmacological traits, we would not suggest that this work provides a depth of information about genetic susceptibility to ALL other than this observation of the potential involvement of gene regulatory defects via allele-specific effects we observed using MPRA. Please see our **reply to Reviewer #1, Comment #7** above.

Reviewer #3 (Remarks to the Author):

The goal of the study by Bhattarai et al. was to test the premise that the underlying non-coding genetic variations and gene regulatory factors would impact the diverse pharmacological traits found in ALL. Based on prior information and GWAS analysis, SNVs were identified and selected for this study. Therefore, the authors tested their interrogation of GWAS regulatory variants that map accessible chromatin sites in ALL by ATAC seq analysis and use MPRA to narrow down the scope of non-coding variants to test. Some of which are believed to be associated with ex vivo chemotherapeutic drug resistance in primary ALL cells from patients and/or ALL treatment outcome (i.e., relapse and persistence of MRD) from patient info. The authors then attempt to correlate these results with promoter ChIP to identify candidate target genes of functional regulatory variants with significant allele-specific effects on reporter gene expression. Lastly, the authors then investigate the impact of the top regulatory variant on transcription factor binding, neighboring gene expression and chemotherapeutic drug resistance in ALL cell line cultures. Regarding the impact, the authors claim that this study represents the largest functional investigation of regulatory variants impacting the pharmacogenomics of chemotherapy treatment and fills an unmet need for large-scale functional examinations of regulatory GWAS variants associated with pharmacological resistance. There are some modest doubts about that claim.

Generally noted on the positive side of the study, the flow of the manuscript appears logical and straightforward to follow. Moreover, the authors make the reader aware of the discrepancies found between the screening results and functional validation of the data provided. The authors do provide some moderate insight from the functional screen of the GWAS SNVs lists and appear fairly comprehensive, with some limitations noted. Generally, the biochemical studies appear somewhat robust with some questions noted below. The authors make efforts to confirm the PU1 immunoprecipitation studies, chromatin IPs, and biotin-DNA pull down, where some effort went into confirming the interaction of PU.1 and its association with the SNV region. In the sense that the authors have been cautious in interpreting their results. Nonetheless, a marginal issue is with the nature of the sequence depth obtained and data analyzed, why not evaluate some additional SNVs to affirm the strength and statistical power of MPRA screening? I believe this maybe be advantageous to stress the strength of the authors' approach.

Despite the strengths noted, some clarity is definitely required to make some sense of the author's datasets used. For instance, all the patient studies (drug-testing, DNA-GWAS, RNA, ATAC) was done by their core. Moreover, data was not released or available for this manuscript review (at least this reviewer couldn't retrieve or identify this information). This is exemplified in Figure 1A, where is unclear the authors' relationship to this dataset a, and I do not see any information about this aspect aside from some sample numbers provided. Additionally, many of the methods used were vaguely and not detailed sufficiently to really understand the manner in which data might give variation and discrepancy in their results This maybe remedied merely by providing additional supplementary information and data to support the authors' description of their functional assays. For instance, what were the quality control (QC) metrics even basic QC metrics were not clear since the authors are mentioning discrepancy in their results. Such as their reporter assay (MPRA and luciferase assays) and their functional validation by CRISPR for instance. I believe readers would like to be aware how and why discrepancies in their data may have arisen. Of course, use of reporter assays, which does not reflect the margins of a promoter used or its native context could be one reason (for instance, where a CRISPR deletion cannot necessarily equate to an actual SNVs, see below detail), but if the expected result is in the opposite direction, It questions how confident readers would be for the authors to apply the MPRA as a screening strategy.

Specific Comments:

1. Figure 3C and within the discussion the authors mentioned ATAC seq results from patients and primary cells. This appears to differ significantly between them. One wonders if that's due to a quality issue between cell lines and primary cells. If the authors think differences arise from cell line vs primary cell, then fresh/frozen primary ALL should be very similar. The authors should perform QC and define their correlation. As mentioned above authors should provide some QC data and their metrics to show all the ATAC result (TSS enrichment and FRIP scores, etc.) are at comparable levels and their stats are reasonable, otherwise, direct normalized counts comparison in Fig S4 would not be meaningful whereas the variable regression of the data would be needed. This reader noticed that they are using FAST-ATAC for all of their fresh/frozen primary and cell lines. This appears as an OK method for cryopreserved frozen samples, but for fresh samples, especially cell lines, this doesn't seem suitable. Lacking the cell lysis step for fresh healthy cell line cells may result in permeabilization and therefore a lack of proper Tn5 tagmentation, which essentially could explain lack of their results confirming the cell line association with the SNVs.

We have now performed TSS enrichment analyses for all ATAC-seq datasets as outlined by the ENCODE consortium in their standards (<https://www.encodeproject.org/data-standards/terms/>) using the "ATACseqQC" bioconductor package (see **Figure 2A** below). Overall, the Reviewer is correct that there is a difference in average TSS enrichment scores across the diverse cell types (see **Figure 2A** below; also see **Supplementary Figure 1a** of the revised manuscript), with fresh samples (primary, cell lines and xenografts) harboring on average lower TSS enrichment scores compared to frozen primary ALL cell samples. However, there is also considerable overlap in TSS enrichment scores among primary ALL cells (fresh and frozen) and ALL cell lines, as well as a large range of TSS enrichment scores within each cell type. These observations suggest that other factors also contribute to data quality. We further identified differences in average peak number among the different cell types, with fresh ALL cells harboring the largest average peak number (see **Figure 2B** below; also see **Supplementary Figure 1b** of the revised manuscript). Although frozen primary ALL cells harbor the highest average TSS enrichment scores but do not have the highest peak number (compare **Figure 2A** with **2B**), overall, we did find a level of correlation between TSS enrichment score and peak number (see **Figure 2C** below).

Although data quality could drive some of the differences in SNV mapping to accessible chromatin sites among cell samples, we strongly suspect the substantially larger number of primary ALL cells (n=144 combined; fresh n=120, frozen n=24) compared to ALL cell lines (n=14) has a large impact on SNV mapping as the primary cell cohort better captures and encompasses ALL diversity and heterogeneity. Indeed, since ATAC-seq (and other functional genomic assays) largely provide a subset of the total number of accessible chromatin sites present (i.e., not all accessible chromatin sites or regulatory elements are identified by a single ATAC-seq experiment), the larger sample size of primary ALL cells will undoubtedly identify more sites, and therefore provide more instances that an SNV will overlap an accessible chromatin site. This notion is supported by the total number of peak summits identified among ALL cell samples (see **Figure 2D** below; also see **Supplementary Figure 1c** of the revised manuscript), with fresh ALL cell samples identifying ~10X more peak summits compared to cell lines.

Importantly, issues surrounding differences in ATAC-seq data quality do not impact our MPRA results, the primary focus of this work, or our main conclusions. Rather, they solely limit the number of variants that can be functionally evaluated in ALL cell lines, and we were able to validate our top hit in Nalm6 (and now also SUPB15 cells; see our **reply to Reviewer #4, Comment #13** below). Moreover, the ATAC-seq data are not a main focus of this study and the vast majority of ATAC-seq datasets (all PDXs, primary ALL cells and 6/7 B-ALL cell lines) were from published studies from our laboratory (e.g., Diedrich et al. *Leukemia* 2021, PMID: 33714976; Barnett et al. *Cell Genomics* 2023, PMID: 38116118). Independent

groups can also integrate our MPRA results with their own ATAC-seq data (or other relevant functional genomic data, e.g., H3K27ac ChIP-seq) in ALL cell samples and perform independent validation studies of identified functional variants from MPRA. In regard to our analysis in **Figure S4 (Supplementary Figure 6** in revised manuscript), we have now validated allele-specific effects on PU.1 occupancy and chromatin accessibility using CRISPR-mediated allele swaps/knock-in in Nalm6 cells (see our **reply to Reviewer #4, Comment #9**).

Figure 2

2. Figure 2: A question to consider is whether the authors could do primary cell ALL MPRA experiments, the transfection and RNA harvest are done with in 24h. Understandably, use of ALL primary cells maybe challenging to perform MPRA, but it has been performed with other primary cancer lines and certainly primary lymphoblasts. It would be very attractive to readers to appreciate how ALL primary cell and cell lines compare, as illustrated in Figure 2D. If the cell line is the culprit for some of this discrepancy, then it would be definitely encouraging to see that some primary cell or patient-derived cell MPRA data exhibiting the same alt/ref expression changes. Expanding MPRA to a broader sample type (not just cell line) would be a nice leap forward for other disease-type screening.

We have now performed MPRA using patient-derived xenografts of primary ALL cells (ALL PDXs) from two patients. We utilized freshly harvested ALL PDXs because primary ALL cells directly obtained from patients are limited, and the cell numbers required for MPRA made experimentation with primary ALL cells obtained directly from patients impossible. The Reviewer is correct that these experiments were technically challenging as primary cells do not grow in culture and are notoriously difficult to transfect. Indeed, to the best of our knowledge, MPRA has not been performed in ALL PDX cell models. Moreover, heterogeneity across diverse PDXs, limited cell numbers and potential changes in the heterogeneous population of cells that successfully engrafts in independent mice also makes identification of optimized transfection conditions challenging. Nonetheless, we performed MPRA in two ALL PDXs (see **Figure 3** below; also see **Supplementary Figure 2b-c** and **Supplementary Table 3** of the revised manuscript). We identified 27 and 67 significant variants with allele-specific effects using MPRA in these two ALL PDXs, including our top variant hit rs1247117. Notably, ALL PDX MPRA data was significantly correlated with ALL cell line MPRA data, with the vast majority ($\geq 72\%$) of primary cell MPRA variants harboring consistent allele-specific effects (**Figure 3** below; also see **Supplementary Figure 2b-c** and

Supplementary Table 3 of revised manuscript). Overall, the ALL PDX MPRA data further validate our observations using MPRA in ALL cell lines.

Figure 3

3. Figure 3b, It is not clear to this reader where the non-cell line data is coming from, (??). In the methods section, MPRA was described as only being tested in ten cell lines. Please provide this information. In Figure 3c, since the authors may want to show the shared SNVs in open chromatin between different sample types. Maybe authors should consider an upset plot?

We apologize for the confusing labeling. These labels in **Figure 3b** (now **Figure 3c**) refer to the 210 MPRA SNVs found in open chromatin in cell lines (in blue) versus the 346 MPRA SNVs found in open chromatin only in non-cell line samples, such as PDXs or primary ALL patient samples (in black). We have adjusted the labeling and figure legend to better support the understanding of **Figure 3**. We have made **Figure 3c** (now **Figure 3d**) an upset plot as suggested.

4. Figure 4 a-c, It maybe beneficial to see how the non-promoter-associated SNV performed in panel “b” as well, since this is the largest group. In panel “b”, the log₂ difference from 2 groups does not look drastically different, even the p value challenges significance. It is worth some explanation for the reader to understand better.

We apologize if this is due to an incomplete explanation of our rationale. In this figure we are focusing on variants that we can confidently associate with a promoter. Our logic here is that it is less rigorous to pursue variants that we cannot definitively associate with a promoter using the promoter ChIC data we have generated. Having a direct association allows us to pursue a mechanistic regulation of a gene in a hypothesis driven manner, whereas if we are investigating a variant for which we have not promoter association we are left to guess which, if any, neighboring genes may be involved with little evidence other than genomic proximity, which can be inaccurate (e.g., Smemo et al. *Nature* 2014; PMID: 24646999). In essence, promoter ChIC was used to further screen and focus attention to high-confidence variant hits with predicted target genes. We have now included an analysis of non-promoter associated SNVs in panel b (see **Figure 4b** of the revised manuscript).

The significant p-value in **Figure 4b** provides statistical evidence that the two distributions are distinct. Overall, only a small subset of variants had exceedingly large allele-specific effects from the MPRA ($\log_2 > 1$), which may contribute to difficulties in interpretation or expectations for drastic differences. Nonetheless, although the overall distributions do not look drastically different, the non-promoter associated hits appear to contain more outliers and also have an average \log_2 allele-specific effect (i.e., magnitude of change) that is larger compared to promoter-associated hits.

5. Figure 4 e, it would increase impact and the strength of the approach if authors could test a few additional top SNVs from their functional assay. Since the authors already have the HiC information this may further strengthen their data and approach.

Each functional investigation involves considerable experimentation, time and effort. As a result, we elected to originally focus on our original top variant hit, rs1247117. Nonetheless, we have now performed a functional validation of two additional top variants, rs7426865 and rs12660691. Please see our **reply to Reviewer #2, Comment #7** above and **Supplementary Figures 8-9** of the revised manuscript for more details. We have also performed an analysis of our original top hit, rs1247117, in SUPB15 cells as requested by **Reviewer #4, Comment #13**, below.

6. Figure 5 a: This maybe an issue for presentation but the IGV tracks should provide a normalized count read across the tracks for each of the tracks shown. It is hard to appreciate the values and levels of enrichment. For instance are they [0-10], [0-100], etc.. It would be important to have this available for readers.

Normalized read count values are most relevant when comparing two datasets of the same experiment/protein to determine differences. However, including the count range on each graph is beneficial for the reader, so we have included this in the revised manuscript (see **Figure 5a** of revised manuscript). Notably, for each experiment shown (ATAC-seq, PU.1, RNA Pol2 and H3K27Ac) there are statistically significant enrichments called at the rs1247117 locus. We note in the manuscript that the variant lies within accessible chromatin harboring binding sites and/or enrichment for the diverse factors/chromatin states described in the **Figure 5a**. These datasets are included in the NCBI GEO accession provided.

7. Figure 5 e-g: One question. Why did the authors delete such a large genomic regions for the functional assay? Couldn't they perform a CRISPR-knockin to introduce the SNV? Such would introduce significant artifacts for chromatin structure, chromatin accessibility, and TF binding. These finding are likely very different by deleting almost 1kb region. Reasoning is not clear to this reader. Moreover, this may also explain why this functional validation does not match their own reporter assays.

We agree that we did not sufficiently explain our reasoning for not pursuing the CRISPR knock-in and instead elected to perform deletions in heterogeneous cell pools. The problem with the CRISPR knock-in is that it is well documented that clonal selection after CRISPR knock-in leads to the accumulation of new variants and even larger structural alterations (Panda et al. *The CRISPR Journal* 2023, PMID: 37071670). Consequently, we cannot be sure if any of the phenotypic changes we see after generating clones are due to the 1-bp SNV we intended to introduce or the other off-target genetic alterations that have arisen through clonal selection. This is particularly problematic for knock-ins that introduce a single SNV within a non-coding element (as compared to knockouts/knock-ins at genes that are likely to result in more extreme phenotypic consequences) and for assessing effects on complex polygenic traits as the effect is likely to be more subtle and could therefore easily masked by off-target effects from clonal selection. CRISPR knock-ins of non-coding elements would be more appropriate for assessing *cis* effects

at the modified locus directly, including effects on the putative target gene (see our reply to **Reviewer #4, Comment #9**). These locus-restrictive studies using knock-ins (included in the revised manuscript) are consistent with our 1-kb deletion analyses, and therefore support our original methodology. In addition to these challenges, selecting the appropriate control following clonal selection is difficult as cell lines are heterogeneous. To illustrate this point, we performed vincristine cell viability analysis using 3 WT Nalm6 cell clones below (see **Figure 4** below). Overall, the independent clones show substantial differences in their sensitivity to vincristine even without any CRISPR alteration. Because of these diverse challenges, we specifically elected to perform CRISPR deletions in heterogeneous cell pools (without the requirement of clonal selection) for our functional validations. We have clarified this point in the revised manuscript.

Figure 4

8. Figure 5h, The authors need to show the zero (0) -time point to show the same viability to start off with, they also need a DMSO control (or just vehicle) group for both WT and DEL samples just to provide base line viability for both. Hence the reasoning is whether the authors truly statistically account for the significance difference between groups where it appears small for such significant **** $p < 0.0001$ values noted from $n=3$. Please clarify.

We apologize for not being clearer about the use of controls for original **Figure 5h**, now **Figure 6e-g**. The untreated cells maintain ~100% viability and therefore each dose and timepoint is expressed as being relative to the untreated matching cell type's viability. In the original graph, the y-axis is labelled "% Viability relative to untreated". The untreated is the vehicle control, and we have label it as such in the revised manuscript. The vehicle for vincristine drug in this experiment is the same RPMI 1640 medium that the cells are grown in. We have now re-analyzed the data as a dose-response curve using non-linear regression curve and identified significant differences between deletion and parental cells ($p < 0.0001$ for the response to VCR at each time point; see our reply to **Reviewer #2, Comment #6** and **Figure 1** above).

Reviewer #4 (Remarks to the Author): expertise in HiC analysis

NCOMMS-23-06280A

The manuscript by Bhattarai, Mobley et al. provides a functional investigation of noncoding variants identified in GWAS studies and ex-vivo drug resistance assays. The approach to validate these putative CRE elements was to perform an MPRA assay on 1696 SNVs and evaluate which of them altered reporter expression. To further prioritize SNVs they combined their results with chromatin accessibility data from primary samples and cell lines, with the assumption that functional elements should be characterized by open chromatin. To identify potential targets of these putative CREs they performed promoter Capture-C and checked which of them were associated with analyzed promoters. Finally, they focused on a specific SNV (rs1247117) and performed molecular and biochemical assay to evaluate its role on transcriptional regulation and vincristine chemoresistance. Despite this approach being interesting and innovative, there are major concerns with some aspects of this work, which may hamper the publication on a high impact journal as Nature Communications. The authors should thus address the following points:

In line 186 the authors wrote “Overall, these data suggest that the chemotherapeutic drug sensitivity and patient treatment outcome SNVs tested were heavily enriched for functional regulatory variants with the potential to impact gene regulation”. This conclusion is too strong based on the data presented:

1. MPRA experiment was performed using cell lines, while most detected variants were identified from primary patients. The functional relevance of these elements could thus be biased by the homogeneity of the cell lines populations, for instance in terms of transcription factors expression. To claim that the experiment enriched for disease relevant variants it would be necessary to perform the assay on primary ALL derived cultures, which better express intra-tumoral heterogeneity.

We agree that inclusion of MPRA in primary ALL cells would strengthen the manuscript. We originally elected to perform MPRA in cell lines as experimentations in primary ALL cells are notoriously difficult. Primary ALL cells do not expand and die *ex vivo* in culture. Moreover, in addition to these logistic challenges, primary ALL cells are difficult to transfect. Despite these technical challenges, in the revised manuscript we have now performed MPRA using patient-derived xenografts of primary ALL cells from two patients (ALL PDXs). We found that our ALL PDX MPRA data was significantly correlated with ALL cell line MPRA data, with the vast majority ($\geq 72\%$) of primary cell MPRA variants harboring consistent allele specific effects (see our **Reply to Reviewer #3, Comment #2** and **Figure 3** above; also see **Supplementary Figure 2b-c** and **Supplementary Table 3** of revised manuscript). Collectively, our ALL PDX MPRA results serve as validation of our observations using MPRA in ALL cell lines.

2. The robustness of the results across the cell lines tested is not so evident from the figures. In fig.2c B-ALL lines manifest a high degree of variability regarding the number of SNVs with differential activity. How do the authors explain that? T-ALL lines seem more uniform. Does this reflect differences in terms of transcription factors expression? Did the authors perform a motif analysis of the candidate sequences used for the assay?

The differences in robustness of the MPRA between cell lines is largely due to differences in transfection efficiency (i.e., technical effects) and not the result of biological effects. Indeed, RS411 and SEM cells are representative of KMT2A-rearranged ALL yet harbor substantially different numbers of functional variants identified from MPRA. It is our experience that some cell lines are more readily transfected than

others. Indeed, primary ALL cells (e.g., ALL PDXs) are difficult to transfect and do not propagate *ex vivo*, and we therefore also see lower SNVs called in these cell models. Overall, the T-ALL cell lines we utilized have superior transfection efficiencies.

3. In fig.2d and suppl.1, correlation plots are centered with higher density around (0;0), suggesting that most pairwise comparison occurred with a log2FC of 0 in both cell lines A and B. This suggests that statistical significance is mainly driven by concordant minimal changes in activity between Alt/Ref and does not really support the idea that these SNVs had a biological relevance.

We apologize for the confusion about **Figure 2d** and any connection to biological relevance. In the manuscript we did not intend to suggest that **Figure 2d** supports the biological significance of these variants. **Figure 2d** is a QC metric showing concordance and reproducibility of the MPRA data reaching the adj. p-value < 0.05 cutoff. In the legend we stated that we have only plotted the data with significant adjusted p-values, and therefore we have not plotted any with log2FC values of 0 in both cell lines. The smooth scatter plot does not have the resolution to clearly demonstrate this. In **Supplementary Figure 1 (Supplementary Figure 2a of revised manuscript)** we plotted all the data to be transparent.

The Reviewer is correct that most significant allele-specific effects have a minor effect on transcriptional output (log2FC <1). Nonetheless, it is difficult to assign biological relevance as MPRA are performed using a non-native promoter in an episomal context and not endogenously within the corresponding genomic locus. These data therefore simply demonstrate gene regulatory activity in this artificial construct. Moreover, it is difficult to predict biological relevance based solely on gene regulatory activity as even small changes to the expression of certain genes could potentially lead to phenotypic effects. Collectively, these data simply demonstrate significant allele-specific effects on gene regulation, independent of any association with biological relevance. We have now clarified the wording to better reflect that the MPRA detected variants with the potential to impact gene regulation.

4. The authors should provide the RNA counts for each SNVs to assess the relevance of the log2 fold changes.

We have provided the MPRA RNA count data in NCBI GEO accession.

5. It would thus be better to modify the conclusion from line 186 and present the data just as the workflow that allowed the prioritization of the variants that were further investigated.

We agree with the Reviewer and now have revised our statement.

6. The frequency of concordance across the cell lines should be clarified, plotting how many SNVs were concordant in 1, 2, 3, .. 10 cell lines in a frequency distribution. It would help to strengthen the conclusions.

We appreciate this helpful suggestion and have included this plot in **Figure 3b** of the revised manuscript.

7. It should be better specified in the text that the luciferase assay was performed on only 11 variants. Claiming that these results validate the MPRA experiment and its robustness is somewhat excessive. Also reporting that “over 500 SNVs had reproducible and concordant effects” is excessive, considering that some/many (it is not clear from the data presented) of the variants may be concordant only in 1/3 of the lines tested.

We have altered the statement regarding luciferase assay validation and now solely state that we validated effects on 11 variants. In regard to the 556 SNVs with reproducible and concordant effects, we defined the meaning of “reproducible” and “concordant” in the manuscript, and therefore believe we have been transparent and consistent in our terminology. Moreover, one would not expect many significant hits across all cell lines (or even most cell lines) as differences in cell line biology, transfection efficiencies and/or other technical factors associated with MPRA would impact the ability to detect allele-specific effects. Our primary goal was to perform this assay in a wide variety of cell lines as ALL is a heterogeneous malignancy, and this would offer a more comprehensive analysis. We chose 3 or more cell lines as a reasonable cutoff considering the complexity of MPRA and heterogeneity among ALL cell lines. To be even more transparent, we have now included in **Supplementary Table 4** that each variant listed had concordant and reproducible allele-specific effects in ≥ 3 ALL cell lines.

8. The results presented in fig.4a provide evidence that most of the variants that were prioritized were not nearby a promoter or connected to a distal one. These data highlight an evident lack of discovery power of the promoter Capture-C approach. It would be important to either analyze publicly available or perform Hi-C or H3K27ac HiChIP experiment to collect more CRE interaction data. It is possible that many of the 136 “non-promoter associated” variants also not associated to distal promoters are associated with other enhancers, directly connected to a promoter. It would be important to address this point, because if these 136 elements actually do not result connected even to other enhancers, they may represent false positive hits of the MPRA assay. In that case, this information would be necessary to evaluate the relevance of this approach.

Many functional genomic assays, including 3D mapping assays, typically can only identify a subset of the total number of regulatory elements or 3D interactions within a cell type in a single experiment. We therefore suspect that at least a subset of variants without promoter connections are false negatives. Indeed, we used rather stringent cutoffs and criteria for our promoter ChiC analysis in order to achieve higher resolution. MPRA hits could be false positives as MPRA utilizes a non-native promoter in an episomal context. Nonetheless, our primary goal using promoter ChiC was to prioritize for those variants that do demonstrate statistically significant, high resolution evidence for promoter interactions for downstream experimentation. Because our investigation centered on single nucleotides/SNVs, higher resolution was more important to us than detecting more interactions at a poorer resolution.

To be more thorough in our analysis of promoter connectivity, we have now performed H3K27ac HiChIP in 8 ALL cell lines (697, BALL1, CEM, Nalm6, RS411, REH, SEM and SUPB15; see **Figure 5** below for a summary of loops identified). These data are included in **Supplementary Figure 5** and **Supplementary Table 6** of the revised manuscript. Despite a larger number of total loops from HiChIP, we identified no additional variants with promoter connectivity using HiChIP. This included direct promoter connections, indirect connections via an intermediate enhancer or even connections with other enhancers.

Figure 5

9. The in vitro assay presented in fig.5d supports the idea the A/G variant has reduced binding affinity for PU.1. However, its functional impact in the endogenous chromatin context is not addressed. It would be important to introduce the variant in a wild type cell line and perform PU.1 ChIP experiments to evaluate changes in PU.1 binding. Alternatively, Nalm6 cells could be genetically modified to harbor two A/A alleles and reproduce the primary data presented in fig. suppl 4c, performing both PU.1 ChIP and ATAC-seq. These in vitro models could also be used to validate the result of fig. 5f-g-h.

We agree that this complementary approach would strengthen our conclusions about allele-specific PU.1 occupancy and chromatin accessibility. We have performed PU.1 ChIP-qPCR, and ATAC-seq in WT/parental Nalm6 cells (rs1247117 A/A) and CRISPR-modified Nalm6 cells (rs1247117 G/G; see **Figure 6A-C** below as well as **Figure 5e-g** of the revised manuscript). These experiments were performed using two CRISPR-modified clones and 3 primers (for ChIP-qPCR; Primer 1= 88bp, Primer 2= 83bp, Primer 3 = 119bp). Overall, we identified greater affinity for PU.1 to the risk G allele (see **Figure 6B** below as well as **Figure 5f** of the revised manuscript). We further identified greater chromatin accessibility using ATAC-seq (see **Figure 6C** below as well as **Figure 5g** of the revised manuscript). Finally, we demonstrate that the G allele leads to reduced EIF3A protein expression (see **Figure 6D** below as well as **Figure 5h** of the revised manuscript). Overall, these data are consistent with our experimentation ablating PU.1 binding via CRISPR-mediated deletion.

Figure 6

The experiments presented in fig.5f-g-h are not sufficient to sustain the biological relevance of the variant analyzed, for key reasons:

10. RNA changes evaluated by qPCR are relatively small, despite being statistical significance. The authors should also provide evidence at protein level for the two genes.

We have performed western blotting showing increased EIF3A protein levels from deletion of the accessible chromatin site spanning rs1247117, which is consistent with our RT-qPCR data (see **Figure 7** below as well as **Figure 6d** of the revised manuscript). Notably, we found CACUL1 protein expression did not change, and the TSS of *CACUL1* was not directly connected to the CRE containing rs1247117. As a result, CACUL1 has been omitted from the revised manuscript for clarity to focus solely on *EIF3A*.

Figure 7

11. The changes in relative viability of cells treated with VCR are relatively small, despite being statistically significant. The authors should also perform other independent assay to evaluate changes in cell proliferation or viability.

We agree that the changes in cell viability are subtle. However, it is difficult to speculate how this effect would translate *in vivo*, in primary ALL cells and/or the impact it would have on persistence of minimal residual disease (MRD) after induction therapy in ALL patients, which is the clinical phenotype the variant was originally associated with (Yang et al. *JAMA* 2009, PMID:19176441). Nonetheless, we have now used the cell apoptotic Caspase-Glo 3/7 assay as an independent assay to validate our cell viability findings (see **Figure 8** below and **Figure 6h** of the revised manuscript). Collectively, our data show greater apoptosis (i.e., greater activated Caspase) in deletion cells, which is consistent with lower cell viability in these cells (i.e., greater vincristine [VCR] sensitivity) using the original Cell Titer-Glo assay.

Figure 8

12. It would be important to perform similar experiments in cells with ectopic inducible augmented expression of CACUL1 and EIF3A to demonstrate their relevance in VCR resistance.

In our revision manuscript, we focused our efforts on *EIF3A* given no effect on the protein expression of CACUL1 from deletion of the accessible chromatin site spanning rs1247117 (see our reply to **Comment #10** above). We have now over-expressed EIF3A (**Figure 9A** below and **Supplementary Figure 10a** of the revised manuscript). Importantly, EIF3A over-expression generated a phenotypic effect consistent with rs1247117 deletion cells (i.e., greater sensitivity to vincristine; see **Figure 9B** below as well as **Supplementary Figure 10b** of the revised manuscript). Collectively, these data establish EIF3A as the target gene of the association.

Figure 9

13. rs1247117 was suggested to have an impact on MPRA assay in multiple cell lines, but the authors performed validation experiments only on Nalm6. They should extend their findings at least to another cell line to strengthen their conclusions.

We have expanded our analysis of rs1247117 in SUPB15 cells (see **Figure 10** below and **Supplementary Figure 7** of the revised manuscript). We demonstrate higher EIF3A protein expression in rs1247117 deletion cells (see **Figure 10A** below and **Supplementary Figure 7b** of revised manuscript) and greater sensitivity to vincristine (see **Figure 10B** below and **Supplementary Figure 7c** of revised manuscript), which is consistent with our observations in Nalm6 cells.

Figure 10

Minor comments:

14. The authors should provide a clearer list of samples used for analysis in a supplementary table, defining how many B and T-ALLs

We have now included a description of all B- and T-ALLs used for MPRA in **Supplementary Table 10** of the revised manuscript.

15. Suppl.table 2: it is not clear which are the 35 positive control SNVs and the 31 that had a significant allelic effect as indicated in the text.

We have now highlighted cells containing positive control SNVs bright green and used red text for positive control SNVs with significant q-values ($q < 0.05$). See **Supplementary Tables 2** and **4** of the revised manuscript.

16. Fig.3c: colors used for boxes of “Cell Line” and “PDX+Patient sample+Cell line” are difficult to distinguish.

These data are now shown as an upset plot for easier interpretation in **Figure 3d** of the revised manuscript (also see our reply to **Reviewer #3, Comment #3** above).

Reviewer #5 (Remarks to the Author): expertise in GWAS methodology in ALL

This paper represents a substantial amount of work to define a set of inherited genetic variants that are associated with response to treatment in ALL patients. This effort has the potential for high clinical significance in the screening of ALL patients for different treatments. Further, this paper combines an innovative gene-mapping strategy using *ex vivo* drug resistance assays in ALL cells with downstream functional analyses to identify regulatory variants for treatment response.

Since all of the downstream analyses are dependent on defining a set of inherited genetic variants, my review will primarily focus on this stage of the paper.

1. Genotyping for the treatment response GWAS was done using DNA obtained primary ALL cells. However, since they are using cancer cells there is some concern that some mutations will be somatic. These are primarily expected to be large indels/CNVs, but there may be some point mutations as well. This caveat of somatic mutations should either be covered in the Discussion or methods used to exclude these should be included in the Methods.

The GWAS variants associated with clinical phenotypes (persistence of MRD and relapse) were genotyped using normal (non-tumor) cells and are therefore germline. This includes our top hit, rs1247117. However, the Reviewer is correct that for the *ex vivo* drug resistance variants, because genotyping was performed using DNA harvested directly from primary ALL cells, we cannot exclude the possibility that some of these variants are somatic and not germline. Distinguishing between the two would require genotyping of non-tumor cells, which is not feasible as non-tumor cells are not available for analysis and the clinical trials associated with this patient cohort has already completed. We have now included in the **Discussion** section that we cannot exclude the possibility that some of the identified variants associated with *ex vivo* drug resistance are somatic.

2. No information as to the number of variants included in the treatment response GWAS or the specific quality control steps applied to the genotype data were included in the Methods except for call rate and MAF.

We have now revised our **Methods** section to include the total number of variants for the treatment response SNVs chosen as well as included additional quality control for SNP genotyping. Please also note that genotyping data was obtained from a previous study (Autry et al. *Nature Cancer* 2020; PMID: 32885175), and therefore consistent quality control and filtering was used as this previous study. We have also included all variants (LD and sentinel) and phenotype (ALL susceptibility/control, MRD, relapse, *ex vivo* drug resistance) included in our MPRA in **Supplemental Table 1**.

3. It appears as if all SNVs with $p < 0.05$ were included from the treatment GWAS, but this threshold seems too liberal even with extensive functional downstream analyses. By combing data from Supplemental Files 2 and 3, it appears as if the 54 SNVs that have reproducible transcriptional effects also have highly significant associations with one or more treatment response it is not clear why such a liberal p-value threshold was used. It would be helpful to present these association statistics for the 54 variants in one table. It would be interesting to see how many of these 53 variants would have been selected if a more stringent p-value threshold was employed.

The use of a $p < 0.05$ threshold was used for two main reasons:

- (1) The GWASs performed were severely limited in power because of small population sizes. Unlike GWAS for disease susceptibility, which typically utilize genotyping data from thousands or even

tens of thousands of individuals, these GWASs were performed using genotyping data from a smaller cohort of patients on clinical trials (for published studies on MRD and relapse) or using primary ALL cells from patients on clinical trials (*ex vivo* drug sensitivity studies). The use of a stringent p-value cutoff would effectively remove the vast majority of these variants from analysis.

- (2) MPRA offers the ability to test 1000s of variants in a massively parallel and high-throughput manner, and we therefore elected to pursue a larger set of variants using a liberal p-value cutoff to be as comprehensive as possible in our analysis.

To combat challenges with GWAS power and to further enrich for functional variants impacting ALL cell biology we further:

- (1) Tested only variants that map to accessible chromatin sites using a large and comprehensive cohort of 161 ALL cell samples as these variants reside in putative functional/regulatory regions of the genome.
- (2) Tested variants that were top hits in published GWAS studies for MRD and relapse.
- (3) For MRD and relapse variants listed only in supplemental tables and/or those that did not meet p-value cutoffs or that were not replicated, we tested only those variants that were independently associated (using separate GWAS analyses) with *ex vivo* drug sensitivity using a $p < 0.05$ cutoff (using drug sensitivity data in primary ALL cells from patients on Total Therapy XV and XVI protocols that were analyzed separately and combined with a meta-analysis).
- (4) Tested variants associated with *ex vivo* drug sensitivity in primary ALL cells from patients on the Total Therapy XVI protocol and that were also identified as eQTLs using a $p < 0.05$ cutoff in the same patient cohort (i.e., using genotyping and RNA-seq data generated from the same primary ALL cells) or in relevant cell types (i.e., Blood and lymphoblastoid cells from GTEX).

Consequently, to combat limited power, we increased our stringency/criteria by employing independent GWAS analyses for other associated relevant phenotypes, where applicable. We have now included p-values for the primary genetic associations of the 54 variants we identified in **Supplementary Table 6** (Trait_Pval column). We acknowledge that our MPRA analysis only provides information on the allele-specific gene regulatory effects of these variants, and any connection with pharmacological effects would require additional validation, as we have provided and demonstrate with experimentation. Nonetheless, because we were comprehensive and have provided extensive functional data across diverse ALL cell samples for this large set of variants at putative cis-regulatory elements, we believe that our rich dataset can be additionally utilized as an important resource for future GWAS studies on ALL pharmacogenomics. Namely, as future clinical trials are performed and additional GWAS variants are discovered, our dataset can be queried to determine if newly discovered variants harbor functional effects on gene expression. We have now also included this information in the **Discussion** section of the revised manuscript. We have also better clarified our **Methods** for variant inclusion in the revised manuscript.

4. Figure 1 is extremely helpful to understand the overall study design for the selection of SNVs. However, it would be helpful to have the number of SNVs that were contributed from each source (Figure 1b).

We have now added the number of SNVs that were contributed from each source to **Figure 1b** of the revised manuscript.

REVIEWERS' COMMENTS

Reviewer #1 (Remarks to the Author):

This study explores the gene regulatory potential of inherited noncoding variants associated with chemotherapy resistance and acute lymphoblastic leukemia outcome.

This is very interesting paper that adds valuable information on the molecular mechanisms underlying the chemotherapy resistance in acute lymphoblastic leukemia.

The authors have addressed all my concerns and the paper can be published in the current form.

Reviewer #2 (Remarks to the Author):

Thanks to the authors for their thorough consideration of reviewer feedback. I feel my comments have been sufficiently addressed and have no further points to raise.

Reviewer #3 (Remarks to the Author):

The authors have made all revisions, edits and important clarifications to advance their study.

While there remain some very minor details that are unclear to this reader, the revisions are considered satisfactory and there are no further concerns with this manuscript.

Reviewer #4 (Remarks to the Author):

The authors have addressed my concerns and I have no further comments.

Reviewer #5 (Remarks to the Author):

All of my concerns in the initial review have been adequately addressed by the authors.

Many thanks,

Andrew DeWan

RESPONSE TO REVIEWERS' COMMENTS

Reviewer #1 (Remarks to the Author):

This study explores the gene regulatory potential of inherited noncoding variants associated with chemotherapy resistance and acute lymphoblastic leukemia outcome. This is very interesting paper that adds valuable information on the molecular mechanisms underlying the chemotherapy resistance in acute lymphoblastic leukemia. The authors have addressed all my concerns and the paper can be published in the current form.

Reviewer #2 (Remarks to the Author):

Thanks to the authors for their thorough consideration of reviewer feedback. I feel my comments have been sufficiently addressed and have no further points to raise.

Reviewer #3 (Remarks to the Author):

The authors have made all revisions, edits and important clarifications to advance their study. While there remain some very minor details that are unclear to this reader, the revisions are considered satisfactory and there are no further concerns with this manuscript.

Reviewer #4 (Remarks to the Author):

The authors have addressed my concerns and I have no further comments.

Reviewer #5 (Remarks to the Author):

All of my concerns in the initial review have been adequately addressed by the authors.

We thank all the Reviewers for their time and helpful suggestions.